# Identifying Knowledge and Process Gaps from a Systematic Literature Review of Net-Zero Definitions

**Jane Loveday** [1,*] , **Gregory M. Morrison** [1,2] **and David A. Martin** [1]

1   Curtin University Sustainability Policy Institute, School of Design and the Built Environment, Curtin University, Bentley, WA 6102, Australia; g.morrison@westernsydney.edu.au (G.M.M.); david@evolutionroad.com.au (D.A.M.)
2   School of Engineering, Design and Built Environment, Western Sydney University, Penrith, NSW 2751, Australia
*   Correspondence: janehitchins@hotmail.com; Tel.: +61-407-039-958

**Abstract:** The use of the term 'net zero' has rapidly and recently become mainstream but is often not well-defined in the literature. A brief history of the term was researched, followed by a systematic literature review to consider the research question: how have the different net-zero terms been defined in the literature, and do they indicate knowledge or process gaps which identify future research opportunities? Academic research articles were searched for the term 'net zero' and filtered for the term 'definition', resulting in 65 articles. Definitions were analysed according to scale: single-building, community, urban-system, and country-wide scale. The search did not return any definitions concerning country-wide emissions (from agriculture, forestry, large-scale transportation, or industrial and mining processes), a surprising outcome given the emissions impact of these areas. The main knowledge and process gaps were found to be in four areas: governance, design, measurement and verification, and circular framework. A clear net-zero definition is required at the appropriate scale (single-building or urban-system scale), which includes explicit system boundaries and emission scopes, life-cycle energy and greenhouse gas (GHG) emissions and should incorporate a dynamic approach. The scale most likely to achieve net zero is the urban-system scale due to the potential synergies of its interacting elements and energy flows.

**Keywords:** $CO_2$; carbon; greenhouse gas emissions; net-zero-energy building; embodied energy; renewable energy

## 1. Introduction

### 1.1. Context

The term 'net zero' is now commonly used in both the current academic literature and in the public domain, with an exponential rise in use over the past 30 years. As a concept, net zero fits within the third wave of environmentalism, the first being environmental awareness and conservation, the second, sustainable development, and the third being climate action. Net zero has become a call to action to avert the climate crisis. Whilst there is clarity around the definition of net zero on a global scale (viz, the need to balance the greenhouse gas emissions the planet produces, with those which the planet consumes measured over a historically relatively short timeframe [1]), it is difficult to break this down into the lesser definitions required to regulate for daily action. This difficulty is due in part to the complexity of defining the scale on which to measure net zero, and partly due to the inconsistent use of nomenclature.

The research community has an important part to play in solving some of these complex issues to achieve effective climate action—but what should the research address? Are there specific areas or actions that require more attention and that will promote a faster more effective response to the climate emergency? In light of these questions, a systematic literature review (SLR) on net zero has been undertaken.

*1.2. Research Question*

The net-zero concept has been introduced as a way of assisting in reducing greenhouse gas emissions to mitigate global warming. However, based on an initial brief search of the academic literature, research into net-zero concepts seems to be focussed on specific technologies such as renewable energy, e.g., [2], or on single built environment units such as net-zero energy buildings, e.g., [3]. A plethora of net-zero terminology is used both in the vernacular and in the academic literature. This raised the questions, what do all these terms mean and what are the consequences of using such a variety of terminology for the achievement of net zero? What should research focus on to successfully progress the concept of net zero? These questions led to the following research question:

How have the different net-zero terms been defined in the literature, and do they indicate knowledge or process gaps which identify future research opportunities?

*1.3. Definitions*

Within this review, there are many variations on definitions of the subsequent terms; however, unless stated otherwise, the following definitions have been used.

1.    Net-zero emissions

The Intergovernmental Panel on Climate Change (IPCC) definition [1] is as follows:

Net-zero emissions are achieved when anthropogenic emissions of greenhouse gases to the atmosphere are balanced by anthropogenic removals over a specified period. Where multiple greenhouse gases are involved, the quantification of net-zero emissions depends on the climate metric chosen to compare emissions of different gases (such as global warming potential, global temperature change potential, and others, as well as the chosen time horizon).

2.    Greenhouse gases (GHG)

The IPCC definition [1] is as follows:

Greenhouse gases are those gaseous constituents of the atmosphere, both natural and anthropogenic, that absorb and emit radiation at specific wavelengths within the spectrum of terrestrial radiation emitted by the Earth's surface, the atmosphere itself and by clouds. This property causes the greenhouse effect. Water vapour ($H_2O$), carbon dioxide ($CO_2$), nitrous oxide ($N_2O$), methane ($CH_4$) and ozone ($O_3$) are the primary GHGs in the Earth's atmosphere. Moreover, there are a number of entirely human-made GHGs in the atmosphere, such as the halocarbons and other chlorine- and bromine-containing substances, dealt with under the Montreal Protocol. Besides $CO_2$, $N_2O$ and $CH_4$, the Kyoto Protocol deals with the GHGs sulphur hexafluoride ($SF_6$), hydrofluorocarbons (HFCs) and perfluorocarbons (PFCs).

3.    $CO_2$ equivalent ($CO_2$-eq) emission.

The IPCC definition [1] is as follows:

The amount of carbon dioxide ($CO_2$) emission that would cause the same integrated radiative forcing or temperature change, over a given time horizon, as an emitted amount of a GHG or a mixture of GHGs. There are a number of ways to compute such equivalent emissions and choose appropriate time horizons. Most typically, the $CO_2$-equivalent emission is obtained by multiplying the emission of a GHG by its global warming potential (GWP) for a 100 year time horizon. For a mix of GHGs, it is obtained by summing the $CO_2$-equivalent emissions of each gas. $CO_2$-equivalent emission is a common scale for comparing emissions of different GHGs but does not imply equivalence of the corresponding climate change responses. There is generally no connection between $CO_2$-equivalent emissions and resulting $CO_2$-equivalent concentrations.

4.    Regulated energy or load: the energy used due to heating, cooling, hot water, fans, pumps and fixed lighting [4].
5.    Unregulated energy or plug load: the energy used by small appliances operated by building occupants, e.g., TVs, computers, washing machines, etc. [4].

6. Operational energy: the regulated energy used in a building, i.e., for space heating or cooling, fixed lighting, or ventilation [5]. Note that in some cases, operational energy also includes unregulated energy.

7. Embodied energy: the amount of primary energy and renewable energy used in the manufacturing, construction and maintenance of materials or buildings, and to some extent also from their deconstruction and disposal of materials [5,6].

8. Embodied carbon or embodied GWP: the amount of GHG emissions created in the manufacturing, construction and maintenance of materials or buildings, and to some extent also from their deconstruction and disposal of materials [5].

9. Life-cycle assessment (LCA): compilation and evaluation of the inputs, outputs and the potential environmental impacts of a product or service throughout its life cycle. This definition builds on ISO documentation [7,8].

10. Primary energy: the raw fuel which can be burned or combusted in order to produce heat and electricity, including natural gas, propane, fuel oils, wood, coal and refined petroleum products [9]. This is equivalent to non-renewable primary energy [5].

11. Secondary energy: the energy product generated from a raw fuel, such as electricity, steam, hot water and chilled water [9].

In this review, the authors use the term 'net zero' on its own to represent the generic concept of energy or GHG emissions balanced at any scale; however, where a specific metric or scale is discussed, the appropriate words are added to the net-zero term, e.g., net-zero energy buildings, net-zero emissions.

Section 2 explains the methodology of this review and Section 3 provides some background on net zero and its place in history. Section 4 describes the terminology used around net zero, and Section 5 discusses the research opportunities presented by the knowledge and process gaps found in Section 4, whilst Section 6 brings the results and findings together into a conclusion.

## 2. Methodology

The following methodology based on the Preferred Reporting Items for Systematic Reviews and Meta-Analyses (PRISMA) reporting guideline [10] was used to conduct this SLR:

- Identify where to search (which databases);
- Determine the main search concepts and terms based on the research question;
- Determine the inclusion/exclusion criteria;
- Analyse resulting search articles;
- Add and use any other relevant articles found manually (a list of the manually sourced references is given in Supplementary List S1).

### 2.1. Databases

To address the research question around the ambiguity of the different net-zero terms, the literature was interrogated using two relevant databases: Scopus and Web of Science. They were investigated in a pilot scoping study and found suitable to find and extract articles relating to net zero.

### 2.2. Search Terms

To identify any knowledge gaps around commonly used net-zero terms, it was not possible to restrict the literature search through pre-empting what these terms were. An initial investigatory search for net zero in the full text across databases returned thousands of results (11,625 in Scopus alone), which further extended to 17,747 in Scopus when the term 'zero net' was included (a term unexpectedly seen in some articles). A brief consideration of the zero net articles identified many which were irrelevant to the climate change topic—most were physics terms such as zero net torque or zero net momentum. However, there were also many relevant articles and the majority of these articles contained: zero net $CO_2$, zero net carbon, zero net energy building, zero net emission, and zero net

greenhouse. The search string was then modified to restrict articles to these particular zero net terms or just the net-zero term.

To restrict the results to articles targeted towards net zero, the search was constrained to the terms appearing in the title, abstract or keywords (TAK). The authors decided to focus on peer reviewed research articles alone, which further reduced the total number of articles.

Table 1 presents the search strings and number of initial resulting articles. These articles were exported from both Scopus and Web of Science into EndNote X9, a reference management software. When full texts of the articles were required, they were imported into EndNote for annotation and EndNote was used for the citations in this review. The meta-data for the resulting articles were exported from EndNote into a Microsoft Excel spreadsheet with columns of Source ID, Author, Year, Title, Abstract, Author Address, Keywords and Journal type. The main Excel functions of conditional formatting, search and filter were used to assess the data.

**Table 1.** Search string and filtering process.

| Process | Scopus (Number of Articles) | Web of Science (Number of Articles) |
|---|---|---|
| Search string: TITLE-ABS-KEY ("net zero" OR "zero net CO$_2$" OR "zero net carbon" OR "zero net energy building" OR "zero net emission" OR "zero net greenhouse") AND (LIMIT-TO (DOCTYPE, "ar")) [Note Web of Science uses TOPIC instead of TITLE-ABS-KEY] Data downloaded to Endnote X9 and then to Microsoft Excel | 1609 (Search date 15 October 2021) | 1483 (Search date 18 October 2021) |
| Number of unique articles (duplicates removed within and between both databases through visual use of conditional formatting in Excel) | 1606 | 375 |
| Number of irrelevant articles found using Excel filters on the title and abstract | 197 (irrelevant) | 167 (irrelevant) |
| Number of articles with no abstract which were removed | 29 | 19 |
| Number of resulting articles | 1379 | 190 |
| Total number of articles | 1569 | |

Duplicates in the spreadsheet were removed using the conditional formatting function on the authors and title columns in combination with filtering and visual inspection.

*2.3. Inclusion/Exclusion Criteria*

2.3.1. Excluded Articles

Articles which were not relevant to climate change and emission reduction were excluded from the data. Through a filtration process in Excel, articles which did not contain the following words in their title or abstract were excluded: climate; environmental; emission; carbon; warming; decarboni*; "Paris agreement"; greenhouse; renewable; "energy building"; "building energy"; solar; sustainab* (where * represents a wildcard and " " represents a phrase).

Furthermore, the words directly following the first appearance of the term 'net zero' in the title, abstract or keywords were extracted for each article using an Excel search function. These words, as well as the keywords for each article, were checked for relevance to the topic of emissions reduction through subjective assessment. All irrelevant articles were excluded. This final database was used to filter for the different terminology and definitions of net zero. The number of articles found for each database is shown in Table 1.

2.3.2. Filtering for Terminology and Definitions

The words immediately following the net-zero term were examined and used to categorise the wide variation of net-zero terms and its derivations (as found in the literature database). To reduce the double counting of terms which appeared in both the title and the abstract and potentially also in the keywords, the number of each of these terms was

counted first from the extraction of terms in the abstract. These papers were then excluded, and the terms were then counted from the title. These papers were also then excluded, and the number of terms were counted finally from the keyword extraction. A frequency table of these terms was generated (Table 2).

To find the definitions of each of these net-zero terms, the filter function in the Excel database was used and all articles with the term 'definition' present in any of the title, abstract or keywords were extracted. Table 1 shows the number of articles extracted from this process.

### 2.4. Search Process

As net zero is such a prominent and contemporary topic, there were over 1500 articles (Table 1). The filter function in Excel was subsequently used to extract articles relevant to a definition of the top terms.

### 2.5. Limitations

In restricting the search string to those net-zero articles which contained the term 'definition', no articles on large-scale land-use, transportation or industrial and mining processes were retrieved, nor were any articles retrieved on heritage buildings. These areas are crucial in achieving country emissions targets, and the lack of definitions in these areas is concerning. However, the large number of articles on net zero (1569) suggests research may well be occurring in these areas and those definitions may be extracted through use of different search terms to those used in this SLR.

## 3. Background

### 3.1. Science of Net Zero

The latest Working Group I contribution to the IPCC 6th Assessment Report has stated the likely range of human-induced global surface temperatures this decade (2010–2019) relative to 1850–1900 is 1.07 (0.8 to 1.3) °C. This is likely broken down into warming contributions by well-mixed GHGs of 1.0 °C to 2.0 °C and by natural drivers of −0.1 °C to 0.1 °C, and cooling contributions by human induced aerosols of 0.0 °C to 0.8 °C. [11] (A.1.3). Warming of 1.07 °C puts us already perilously close to the Paris Agreement goal of keeping warming well below 2.0 °C, preferably limited to 1.5 °C, compared to pre-industrial levels [8].

To keep warming to a minimum, the Paris agreement urged nations to commit to achieving a balance of GHG emissions, or net-zero GHG emissions, in the second half of this century [12]. Whilst some sectors will be able to achieve zero emissions in the required timeframe, the net in net zero is important as this allows other sectors to offset (or sink) emissions which are difficult to remove.

There are some subtleties to the required reductions. As defined in Section 1.3, GHG emissions comprise $CO_2$ and other gases such as $N_2O$, $CH_4$, and $O_3$. These other gases are relatively short-lived in the atmosphere compared with $CO_2$, which can remain for 300–1000 years [13]. Thus, reducing the longer-lived $CO_2$ emissions faster will reduce the amount of locked-in warming and sea level rise due to the large but delayed and long-term impacts of melting permafrost, glaciers and ice sheets [14]. The shift to cleaner forms of energy to reduce GHG emissions will also reduce aerosol concentrations, leading to further warming (aerosols have a cooling effect in the atmosphere). To compensate for this, it is important to concurrently reduce shorter-lived GHG emissions with larger GWPs. The respective cooling and warming effects of these actions will essentially cancel each other out, after a delay of around a decade, during which slight warming will still occur [8] (Chapter 1).

Following the recent COP26 meeting in Glasgow in November 2021, 10 more countries have announced a net-zero target (ranging from 2045 to 2070 for the G20 countries), adding to the 130 countries who have already pledged, altogether covering 90% of global emissions [15]. However, these targets often vary in several ways, for example, the types of

gases and the metric used for comparison, the timeframe for achieving balance, whether the target is legally binding or not. This makes it difficult to assess and hold parties to account [16] and necessitates a framework for assessing net-zero targets which considers scope, adequacy and fairness, and a long-term roadmap [17].

The setting of long-term emission targets must be complemented with short-term targets to factor in the subtleties discussed above [8]. As part of the Paris Agreement, countries agreed to strengthen their 2030 targets at COP26; however, only 92 countries delivered on this pledge [18].

In terms of emissions accounting, the Greenhouse Gas (GHG) Protocol provides a standardised framework for measuring and managing GHG emissions, where they categorise emissions into Scope 1 (all direct GHG emissions), Scope 2 (indirect emissions from consumption of purchased electricity, heat, or steam) and Scope 3 (other indirect emissions such as the extraction and production of purchased materials and fuels, transport-related activities in vehicles not owned or controlled by the reporting entity, electricity-related activities not covered in Scope 2, outsourced activities, waste disposal, etc.) [19].

### 3.2. The Historical Basis for Net Zero

Based on the original search for net zero and zero net in the full text of articles listed in Scopus, the first use of the term 'net zero' was in 1957 in a physics article about electric fields—net-zero-field mobility [20]. Then followed four decades of usage mostly in physics or physical chemistry, e.g., net-zero phase change, net-zero dipole moment. In 1991, the first climate related use was found to be zero net $CO_2$ emissions, which was mentioned in relation to the use of biomass fuels, the greenhouse effect and planting trees in tropical countries [21]. The first use of climate-relevant net zero in the literature was in 2001, again in relation to biomass burning, and the results from the analysis was that "carbon emission estimates from annual region-wide sources of deforestation and biomass burning in the early 1990s are apparently three to five times higher than reported in previous studies for the Brazilian Legal Amazon", which importantly refuted other studies suggesting that biomass burning provides a net-zero annual source of terrestrial carbon [22].

Stepping outside the literature review research articles, a search for net zero was performed in the reports from the IPCC, as well as a general search of the internet. This was to determine when net zero was first used in the IPCC reports, and to see if the origins of the climate related term could be found.

One website showed a timeline of net zero [12] and listed some relevant scientific articles published in *Nature* in 2009 regarding the significance of cumulative $CO_2$ emissions in determining the future extent of global warming. This could be used as a framework to both assist in mitigation and to determine future warming impacts [23,24].

A media article written in 2019 was found in Climate Home News, which described the history of the term 'net-zero emissions' [25]. The report stated that in 2013 in a country estate house in Scotland, a focus group of around 30 women ranging from lawyers, to diplomats, financiers and activists gathered to discuss the failure of the 2009 Copenhagen COP meeting and the future messaging needed to bring countries together to tackle climate change. Amongst these were Christiana Figueres, head of the UN climate body, Rachel Kyte, climate envoy for the World Bank, and Farhana Yamin, a climate lawyer, author, speaker and activist. Yamin said:

> "When I reflected on why Copenhagen failed, I came to the conclusion ... we needed a more concrete, practical measure".

The narrative continued to relate how some experts were focusing on $CO_2$ emissions concentrations, whilst a separate scientific discourse was considering carbon budgets which would contract and converge to bring the worlds rich and poor on to equal budgets. However, Yamin took inspiration from the Montreal Protocol and advocated for zero emissions, stating:

> "Once people get their heads round this scary idea, they enjoy having this constraint and something to work towards."

The focus group did question whether the net in net zero provided too much wiggle room and might slow the process of emission reduction in favour of later technologies to remove $CO_2$ from the air, but the simplicity and universality of net zero was agreed upon as the best approach. The concept of net zero was then promoted by this informal group of women within their own networks around the world.

The term 'net-zero-energy building' seemed to be the first appearance of net zero in the IPCC reports, appearing in the Building chapter of the 2014 Working group III report [26]. However, the tougher target of net-zero emissions, agreed upon by the focus group, was only recognised in the IPCC Special Report into Global Warming in 2018 [8].

Other contextual world events as well as the above-mentioned reports and articles are shown in Figure 1, with the frequency of net-zero and zero-net articles (found in any of the TAK from the SLR) given by the solid black line.

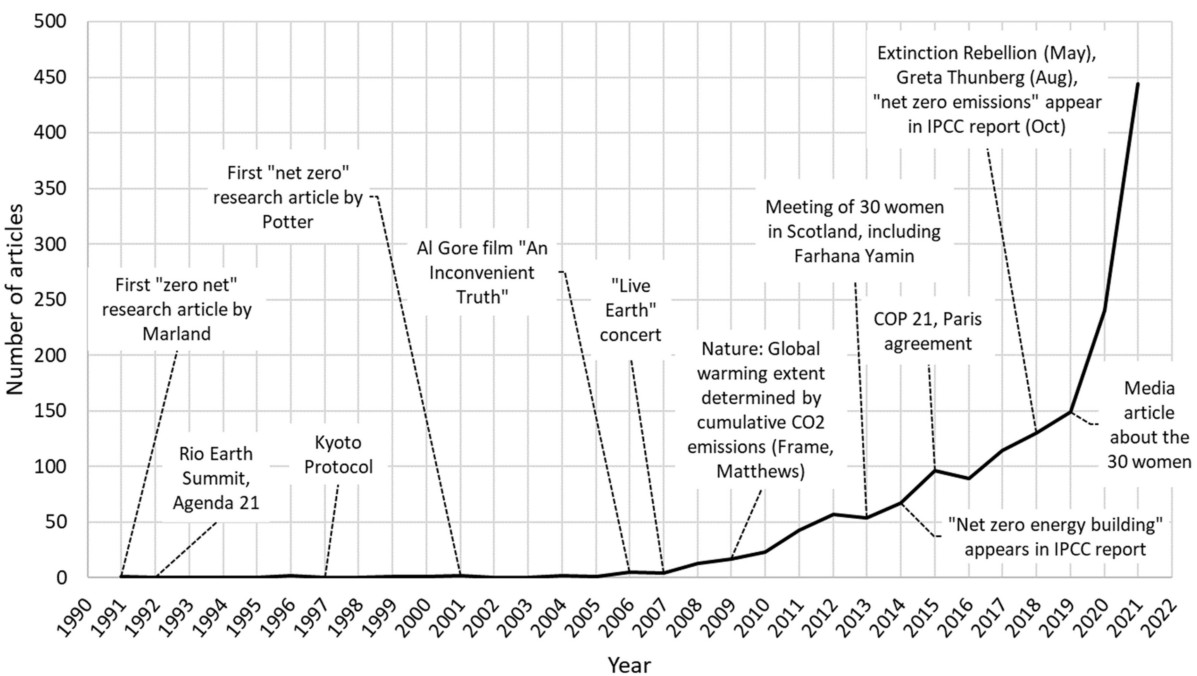

**Figure 1.** A timeline showing the frequency of research articles using the terms 'net zero' or 'zero net' in their title, abstract or keywords, and relevant historical events.

## 4. Terminology Analysis

Technically, the conceptualisation of net zero centres around emissions, whether they are $CO_2$ alone or include all greenhouse gases. However, a search for the word 'emission' in the articles returned fewer numbers than would be expected. The frequency with which the term 'emission' appears either directly next to net zero (such as net-zero emission), or at all, was investigated (Sections 4.1 and 4.2). The other net-zero terms were studied for their definitions, and a theme of scale was identified. The definitions were thus classified according to scale (Section 4.3).

### 4.1. Frequency of the Term 'Emissions'

The number of articles without the word 'emission' appearing in each of the Title, Abstract, Keywords or in any of the Title, Abstract or Keywords (TAK) were:

- Title: 1409 (89.8%)
- Abstract: 855 (54.5%)
- Keywords: 1108 (70.6%)
- Any of the Title, Abstract or Keywords: 809 (51.6%)

Thus, researchers searching for net-zero articles who use the word emission to reduce the number of resultant articles to a manageable number must be aware that they could be discarding a greater proportion of potentially relevant net-zero articles.

It was noted that the number of keywords for articles ranged from 1 to 68. Caution must be used if relying on keyword searches alone, as some keywords are chosen by the content suppliers to the database, and a search of these may return results that do not accurately reflect the content of the article.

The following section examines the terms appearing directly after the term 'net zero'.

### 4.2. Net-Zero Terminology

Table 2 shows the frequency of the terms appearing directly after net zero in any of the TAK. Terms were extracted and counted in the order: abstract, title, and then keywords, as some articles mentioned the net-zero term in more than one of these locations. The frequency was calculated as a percentage of all articles where net zero was first mentioned in the TAK (1472 articles). Note that this process extracted the terms next to the first mention of net zero only, and terms next to further mentions of net zero were not extracted.

**Table 2.** Frequency of terms appearing directly after 'net zero' in the abstract, title or keywords.

| Terms after Net Zero | Number (%) of Net-Zero Articles (Out of 1472) | Terms after Net Zero Cont. | Number (%) of Net-Zero Articles (Out of 1472) |
|---|---|---|---|
| energy building | 370 (25.1) | environmental impact | 4 (0.3) |
| energy | 190 (12.9) | society | 4 (0.3) |
| emissions | 171 (11.6) | carbon energy | 3 (0.2) |
| just "net zero" | 135 (9.2) | carbon homes | 3 (0.2) |
| greenhouse gas | 74 (5.0) | emission energy | 3 (0.2) |
| energy home | 65 (4.4) | energy consumption | 3 (0.2) |
| target | 48 (3.3) | energy development | 3 (0.2) |
| carbon emission | 42 (2.9) | energy neighbourhood | 3 (0.2) |
| $CO_2$ emissions | 38 (2.6) | energy school | 3 (0.2) |
| building | 33 (2.2) | positive energy | 3 (0.2) |
| carbon | 31 (2.1) | urban water | 3 (0.2) |
| emissions target | 25 (1.7) | community | 2 (0.1) |
| energy residential | 18 (1.2) | costs | 2 (0.1) |
| carbon dioxide | 17 (1.2) | data centre | 2 (0.1) |
| water | 17 (1.2) | emissions society | 2 (0.1) |
| energy community | 11 (0.7) | emitters | 2 (0.1) |
| electricity | 9 (0.6) | energy water | 2 (0.1) |
| energy target | 9 (0.6) | energy/emissions | 2 (0.1) |
| carbon footprint | 8 (0.5) | forcing | 2 (0.1) |
| home | 7 (0.5) | fossil fuels | 2 (0.1) |
| exergy | 6 (0.4) | multi-energy | 2 (0.1) |
| carbon economy | 5 (0.3) | school | 2 (0.1) |
| emission building | 5 (0.3) | structure | 2 (0.1) |
| carbon building | 4 (0.3) | transition | 2 (0.1) |
| economy | 4 (0.3) | vision | 2 (0.1) |
| energy district | 4 (0.3) | | |

Similarly, the frequency of the terms appearing directly after 'zero net' in any of the TAK were extracted, again in the order of abstract, title, and then keywords. The frequency was calculated as a percentage of all articles where zero net was first mentioned in the TAK (127 articles). The results are shown in Table 3. Note that the number of articles which mention net zero and zero net do not add up to the total 1569, as 30 articles mention both net zero and zero net.

**Table 3.** Frequency of terms appearing directly after 'zero net'.

| Terms after "Zero Net" | Number (%) of Zero Net Articles (Out of 127) |
|:---:|:---:|
| emissions | 31 (24.4) |
| energy building | 19 (15.0) |
| energy | 15 (11.8) |
| carbon emissions | 14 (11.0) |
| energy homes | 10 (7.9) |
| greenhouse gas | 9 (7.1) |
| $CO_2$ | 4 (3.1) |
| energy consumption | 4 (3.1) |
| carbon | 3 (2.4) |
| $CO_2$ emission | 3 (2.4) |
| electricity | 3 (2.4) |
| energy operation | 2 (1.6) |
| building | 1 (0.8) |
| carbon electrical power | 1 (0.8) |
| carbon energy | 1 (0.8) |
| energy city | 1 (0.8) |
| energy community microgrid | 1 (0.8) |
| energy demand | 1 (0.8) |
| energy standard | 1 (0.8) |
| pollution | 1 (0.8) |
| waste | 1 (0.8) |
| water | 1 (0.8) |

The most common term was 'net-zero-energy building', which was mentioned in close to one quarter of all articles and was twice as frequent as the next term, 'net-zero energy', and the similarly frequent term 'net-zero emissions'. Interestingly, when articles mention 'zero net' instead of 'net zero', the most frequent term was 'zero net emissions' at close to one quarter of all articles. 'Zero net energy buildings' was the second most frequent term (15%) followed by 'zero net energy' (11.8%).

Adding these top three terms for both net zero and zero net together accounted for 796 articles (or approximately 50% of the total number of articles)—there may be some overlap where the article mentions both terms, although this occurs in less than 2% of articles.

When comparing how often the terms 'energy' and 'emissions' occur in the word immediately following net zero, regardless of the fourth word in the term, 'energy' occurs 688 times (46.7%) and 'emissions' occurs 110 times (7.5%). Similarly, for 'zero net', 'energy' occurs 54 times (42.5%) and 'emissions' occurs 31 times (24.4%). Hence, the use of the word 'energy' dominates the net-zero discussion in the literature. When searching for articles relating to net-zero emissions, researchers must be aware of the limited use of the term 'emission' in the research literature.

### 4.3. Definition of Terms

For researchers to be able to compare their results, there needs to be consistency across the definitions of net zero. Definitions for each of the main net-zero terms from Tables 2 and 3 were sought. The original 1569 articles were filtered for the word definition in any part of the title, abstract or keywords, returning 65 results. Of these, 12 did not contain relevant definitions, leaving 53 articles to be studied.

One approach is to consider net zero in terms of system boundaries with application to different scales, from micro to macro scale, i.e., boundaries could encompass a single building, a neighbourhood/community, a city, or even a country [27]. Scale has also been defined by the area or number of buildings, power usage, and who the owner or decision-maker is [28]. In this review, the individual building and community-scale definitions [28] have been used. However, a different classification was derived for the other scales based

on descriptions found in the 53 articles and are mainly based on the typology of the buildings. On the community scale, the building typologies are similar, and end users, e.g., residents, may have similar energy requirements over similar periods. This scale has benefits over the building scale in terms of load diversity, technical integration, and economies of scale [28]. However, more opportunities exist when groups of buildings have different typologies such as warehouses, office buildings and residential housing, which have diverse end-users, and which have different resource requirements at different times. Together, these buildings can take advantage of resource and renewable energy generation sharing, leading to synergies in emission reductions. This scale has been termed the urban-system scale. Whilst there may be energy sharing on the community scale, it is unlikely to provide as many energy benefits as for the urban-system scale. No articles were found which provided definitions on a country-wide scale, which may include large-scale land-use (agriculture, forestry) and transport, or industrial and mining processes. This is a surprising result given the emissions impact of these areas and points towards a research gap. Hence, the categories used in this review are shown in Table 4.

**Table 4.** Definitions of different net-zero scales used in this review.

| Scale | Description |
| --- | --- |
| Building | single building unit/apartment/house |
| Community | Several buildings consisting of the same building typology, e.g., all houses [28], and their interactions |
| Urban system | a complex system consisting of a mixture of different building typologies, e.g., commercial and industrial building(s), and residential housing, and other built environment infrastructure, and their interactions |

The complexity of assessing and designing for net zero increases as the scale is increased and more elements are introduced into the system. There is a lack of consistent definitions across all scales, i.e., many different net-zero terms are used to describe essentially the same system, whilst others using the same terms, are not comparable because they use different system boundaries. There is an urgent need for a consistent definition and assessment rules for the concept of net zero at each of these scales.

4.3.1. Building Scale

Building structures along with energy expended in their construction, maintenance and demolition can account for half of the overall energy consumption of society [29]. In Australia, around one-quarter of the annual carbon emissions come from the construction, maintenance, and use of buildings alone (from [30] as referenced in [28]).

A building is defined here as a stand-alone structure, whether it be a residential home, a commercial building, an apartment building, or a factory. The following terms have been used for this scale (note the leading term net is sometimes dropped from the discussion):

| | | |
| --- | --- | --- |
| just net zero | net zero energy building | net zero life cycle energy |
| net zero building | net zero energy homes | zero carbon building |
| net zero carbon | net zero emissions | zero emission buildings |
| net zero carbon home/housing | net zero emissions building | zero energy buildings |
| net zero energy | net zero GHG emissions | net zero standard |

There were 44 articles (83% of the 53), relevant to this scale.

*Definition and Distinction between Energy and Carbon*

Energy

Energy is the actual amount of work produced or consumed by a system and is measured in Joules. In this literature, energy is quantified in terms of electrical or heat generation and is measured in kiloWatt hours (kWh).

It is worth noting here that another very common term found in the literature was nearly-zero energy buildings (nZEB). The many definitions of this differ by the value of nearly in terms of the balance of annual energy use per square metre [31,32]. Net-zero energy is equivalent to the nearly being equal to zero.

Most articles relating to net zero on the building scale were those using the term 'net-zero-energy building' (NZEB) (20 out of 44, or 45%). Others simply referred to zero-energy buildings (14 out of 44, or 32%), or just net-zero energy (11 out of 44, or 25%), with some overlaps where multiple terms were used. Net-zero energy first appeared in US law in relation to commercial buildings [33]. This may be why 9 of the 11 articles directly relate to NZEB; thus, net-zero energy is used as a proxy for NZEB. Of the remaining two articles, one did not provide a definition, and the other was the only one with a unique non-building-related use of net-zero energy [34]—this article is included at the urban-system scale (Section 4.3.3). Some articles discussed both net-zero energy and net-zero carbon as if they were equivalent [4,33,35].

In general, the term 'energy' used in the net-zero concept on a building scale was largely associated with the operational energy of a building.

<u>Carbon</u>

The term 'carbon' is not used in a strictly scientific manner and has been referred to in the literature as meaning any of the following: carbon emissions, carbon dioxide ($CO_2$) emissions, carbon dioxide equivalent ($CO_2$-e) emissions or greenhouse gas (GHG) emissions. These are the gases released during the production or consumption of energy and/or materials. A clear definition of what each of these carbon terms mean was not found in the articles. $CO_2$ emissions are quantified by mass, and in the IPCC definition, the units are metric tonnes of $CO_2$ ($tCO_2$) ([36], Table A.II.3). GHG emissions are equivalent to $CO_2$-e emissions and the IPCC define these using GWPs over a 100 year period ([36], Table A.II.3 footnote).

Net-zero carbon was defined in six (14% of) articles related to the building scale. On this scale, it has also been termed 'net-zero carbon homes' or 'net-zero carbon buildings' [35], as well as 'net-zero carbon emissions' [4]. Net-zero carbon has also been associated with a life-cycle analysis or assessment (LCA) and discussed in terms of embodied energy as well as operational energy (reviewed in [27]); it has also been used to consider energy use and generation [37]. It has further been extended to net-zero carbon footprint [38], which is relevant on the urban-system scale and is discussed in Section 4.3.3. The term 'zero-carbon building' (ZCB) was used as a general term by [39] to collectively represent both nearly zero and net-zero energy buildings. It has also been used for a building which is assessed to have net zero, or negative, carbon emissions [4]. There is some confusion about whether articles are defining carbon as $CO_2$ emissions or as $CO_2$-e emissions. In a review of 35 building assessment approaches, [27] found that the most used term was 'zero carbon', even though the frameworks included both $CO_2$ and $CO_2$-e emissions. This failure to use scientifically correct nomenclature leads to a lack of clarity in the literature.

In general, the term 'carbon' used in the net-zero concept on a building scale was associated with emissions, both from the embodied and the operational energy of a building, as either $CO_2$ emissions or $CO_2$-e emissions.

*Definition*

There is a general lack of consensus concerning a common definition for net zero on a building scale. This is mainly with respect to the metric (energy, energy cost, emission, and exergy), domain (site, and source), timeframe (monthly, yearly, and life cycle), supply boundary (onsite, nearby, and off-site), and type of balance (generation/demand and import/export) [40,41]. There are also inconsistencies in the inclusion (or exclusion) of embodied energy or of GHG emissions [3].

Most definitions include:

- Sustainable building design such as the Passivhaus concept (as this reduces building energy demand) [37,42–45].
- Renewable energy generation either on-site or nearby [44].

- Connection of the building to the grid to cater for seasonal fluctuations, as well as buildings with smaller roof areas with lower renewable generation potential [46].
- A balance between weighted demand and supply (source or primary energy) [47].
- Plug loads, which are generally not included in the EU but are included in the US [48].
- An assumption that net-zero balance is over an annual timeframe.

Although the timeframe over which net zero is measured is usually annually, some studies investigated life cycle energy [5,49], whilst another two studies, although audited annually, were based on monthly credits [47,50]. An annual basis is commonly used as, even with net metering, monthly net zero is not generally cost-effective compared with achieving annual net zero, given the increased amount of solar photovoltaics (PV) required in order to be net zero over the winter months—even in a warm climate such as Brazil [51]. Only small buildings (two–three storeys) in warm climates would find the monthly net-zero timeframe to be cost effective.

Numerous articles mentioned the definition of NZEB given by [52] as balancing any of four metrics: site energy, source energy, energy costs, or $CO_2$ emissions [9,33,41,53,54]. Although this article was not discovered in the SLR process as it is a conference paper, it is included here. The article is eligible for inclusion in this SLR as it has been referenced by many others.

The definitions of the four metrics are as follows (summarised in [54]):

- Site energy. The built environment provides at least as much renewable energy as it consumes annually when accounted for at the site, e.g., [55]. This metric has been more commonly used in the US [48].
- Source energy. The built environment provides at least as much renewable energy as it consumes annually when accounted for at the source, e.g., [56]. Source energy refers to the primary energy used to generate and deliver energy to the site. It accounts for all losses which occur during the processes of extraction, conversion, transport and distribution and uses weighting options to calculate the energy balance [9]. The chosen or regulated weighting options can have a significant effect on building energy balance [9]. This metric is more commonly used in the EU [48].
- Energy costs. The owner of the built environment obtains (earns) at least as much money from selling renewable energy as they pay for energy annually. One example of this is where the client (a school) chose this method to achieve zero energy [57]. Net metering provides a powerful incentive to achieve both net-zero energy and net-zero energy costs [9].
- $CO_2$ or energy emissions. The built environment provides at least as much emissions-free renewable energy to compensate for emissions from all the energy consumed by the building annually.

The easiest metrics to implement are site energy and cost [52]. The most representative and the most used metric, at least up until 2013, were source energy and $CO_2$-e emissions [35]. The latter is particularly important as it relates directly to climate change emissions targets, and there has been a shift to focus on net zero GHG emissions buildings as a metric rather than using energy demand as a proxy for the environmental performance of a building [27]. However, current climate goals by policymakers and legal requirements in the building industry are generally still focused on energy performance. It is essential that the focus be shifted towards including GHG emissions as well, to effectively achieve climate targets—the energy goals address the conservation of resources, whilst the GHG emission goals address climate protection. The reduction in GHG emissions in this industry has generally not yet been adopted in legislation [27].

Figure 2 presents a summary of the possible operational processes which may be included in different definitions of a NZEB. More basic definitions just include the building related operations (Definition 1); Definition 2 extends this to include user related operations, while Definition 3 also includes other building-related sources [5].

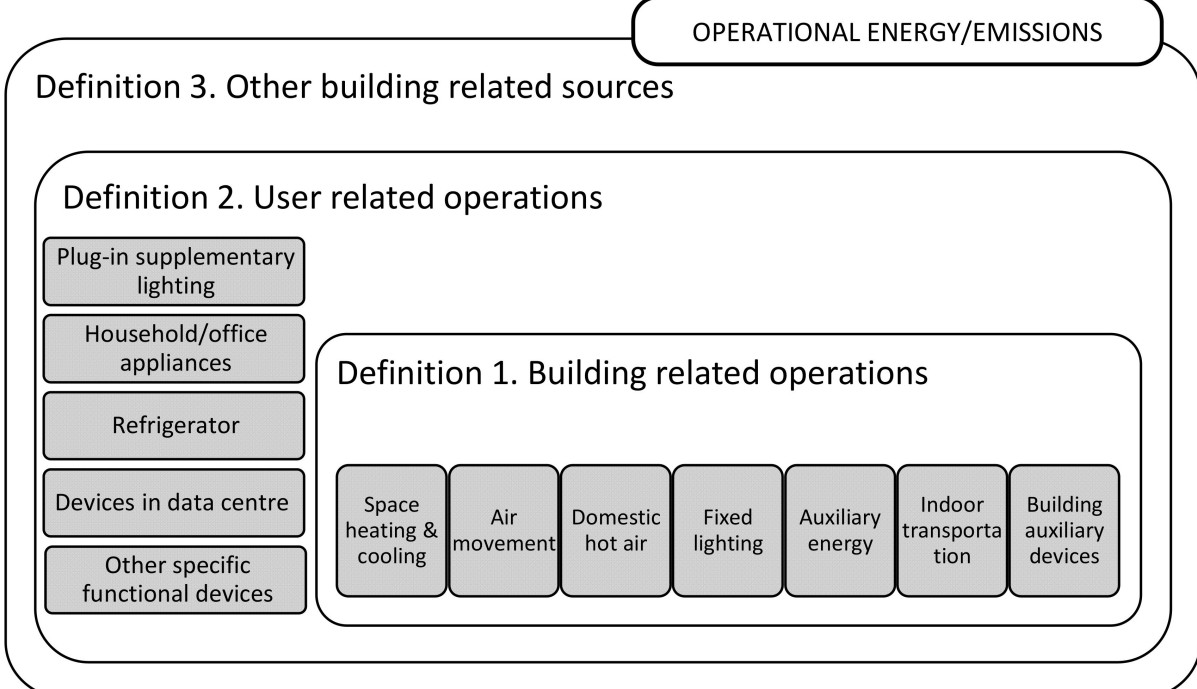

**Figure 2.** Common types of definitions identify the included operation-related items in the energy or emission balance of a NZEB. Adapted and reprinted with permission from [5], Copyright 2015, Taylor & Francis.

*Future Inclusions*

Embodied Energy

Only considering operational energy and not the embodied energy of the building shifts the incentive towards increasing the latter to reduce the former. It does not encourage the elimination of the total energy needs of a building altogether and thus does not aid in the overall reduction of greenhouse gas emissions for the building sector [27,58,59]. Some countries are further advanced in this space than others, as evidenced in a review of zero energy building definitions, where the embodied energy was found to be included only in definitions adopted by Norway and Switzerland [48].

Most building-scale definitions in this review do not include the embodied energy of the building, the on-site renewable energy systems, or energy involved in the transportation of residents [46,50,58,60]. The exceptions to this were four articles whose authors discussed using an LCA concept for the building only: a cradle-to-gate analysis (which included all energy inputs to a product, expressed as primary energy, from extraction to manufacturing, until the product leaves the factory gate, but which did not include construction, building maintenance, or building end-of-use options) [49]; an extension to the life-cycle energy concept through using an emergy (not energy) balance, where solar emergy is the available solar energy previously used-up, both directly and indirectly, to make a service or a product [61]; an analysis of the different system boundaries (modules) which may be used in definitions for NZEB when embodied energy is included (Figure 3) [5]; and a life cycle energy assessment (LCEA), where the embodied energy was found to be a significant component of the energy usage, making up 48.2% and 60% of the total life-cycle energy usage for case studies of single detached residential buildings in Adelaide and Melbourne, respectively [62].

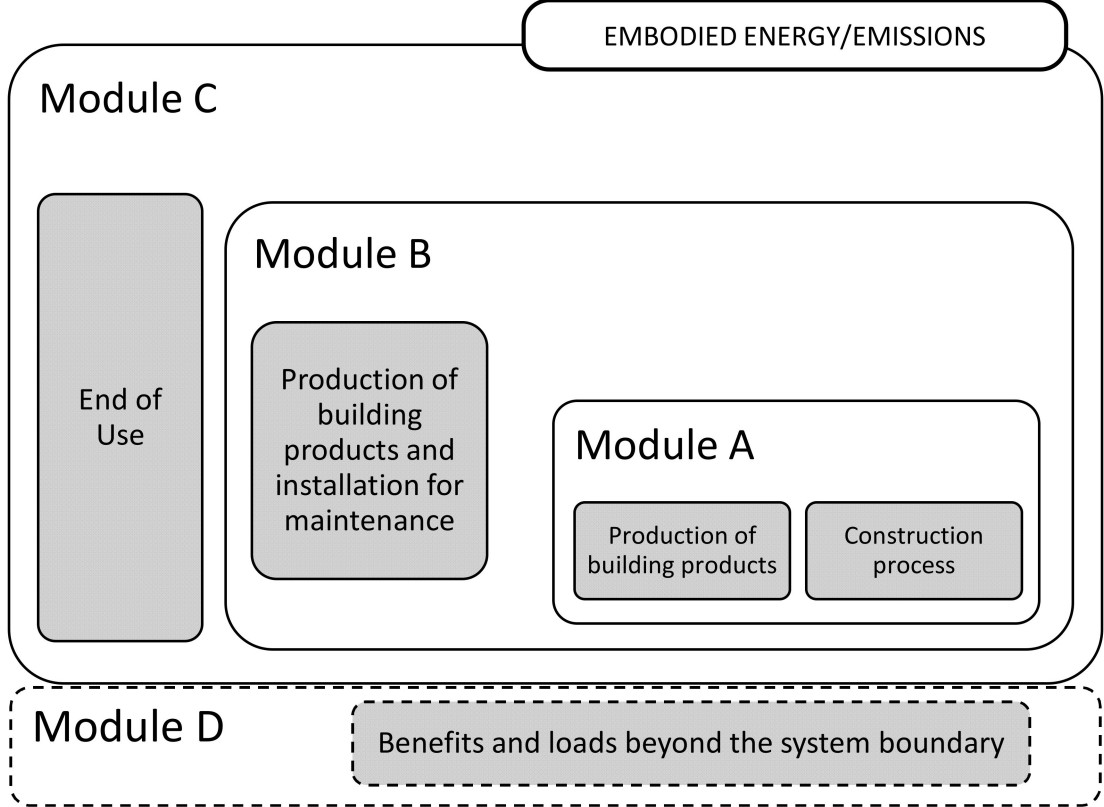

**Figure 3.** Common types of definitions identify the included building-related items in the energy or emission balance of a NZEB. Adapted and reprinted with permission from [5], Copyright 2015, Taylor & Francis.

In a comparison of four sustainable design rating schemes (LEED, Living Building Challenge, Net Zero Energy Building, and Passive House) against an ASHRAE benchmark using a life-cycle assessment framework, it was concluded that focusing on operational energy balance alone (as in NZEB), limits the effectiveness of a scheme to reduce the global warming potential [63]. Introducing and adhering to an embodied energy or GHG emissions budget in the building design process could help address this shortcoming [5].

New policies should adopt both a life cycle approach and a systems approach to avoid shifting energy or emissions from one life cycle stage to another, without an overall reduction [59]. Choosing building materials that lock in the GHG emissions for decades into the future was strongly recommended. The choice of building materials could be based on the suggested development of an international database of embodied environmental flows coefficients using a hybrid analysis approach, which would reduce current underestimation of both embodied energy and emissions [59]. The adoption of an internationally recognised database would allow direct comparison between projects [27,59].

As well as considering the choice of materials, thought must also be given to include the embodied energy of replacement building components, as these can be as significant as the construction-related components, depending on their replacement rate [27]. These rates of replacement may be difficult to determine as they can vary due to policy incentives. For example, the replacement of solar PV systems may occur every 10–15 years once the inverters reach their useful life, despite the panels being warrantied to 30 years [64]. This may occur due to rebate incentives on the installation of new systems, the expiry of the Feed In Tariff period (money paid for excess energy returned to the grid), and the increased efficiency of the latest technology panels, generating more electricity per square metre for the owner than their previous array—in some cases, doubling their capacity [64]. However, future increases in the amount of renewable energy in the energy mix will reduce the

GHG emissions intensity of new building materials and components over time. Hence, a dynamic approach to embodied energy for both replacement components and future builds needs to be employed in an LCA (this is currently being investigated by the IEA EBC Annex 72 [65]) [27].

The system boundary definitions shown in Figure 3 have been further adapted and extended by [27] who present a modular framework for the system boundaries of an LCA for a building (Figure 4). The framework has additional sections B6, B7, and B8 in the operational impacts section which incorporate the more complex analysis of unregulated energy use (plug loads), water use, and transport induced by the building.

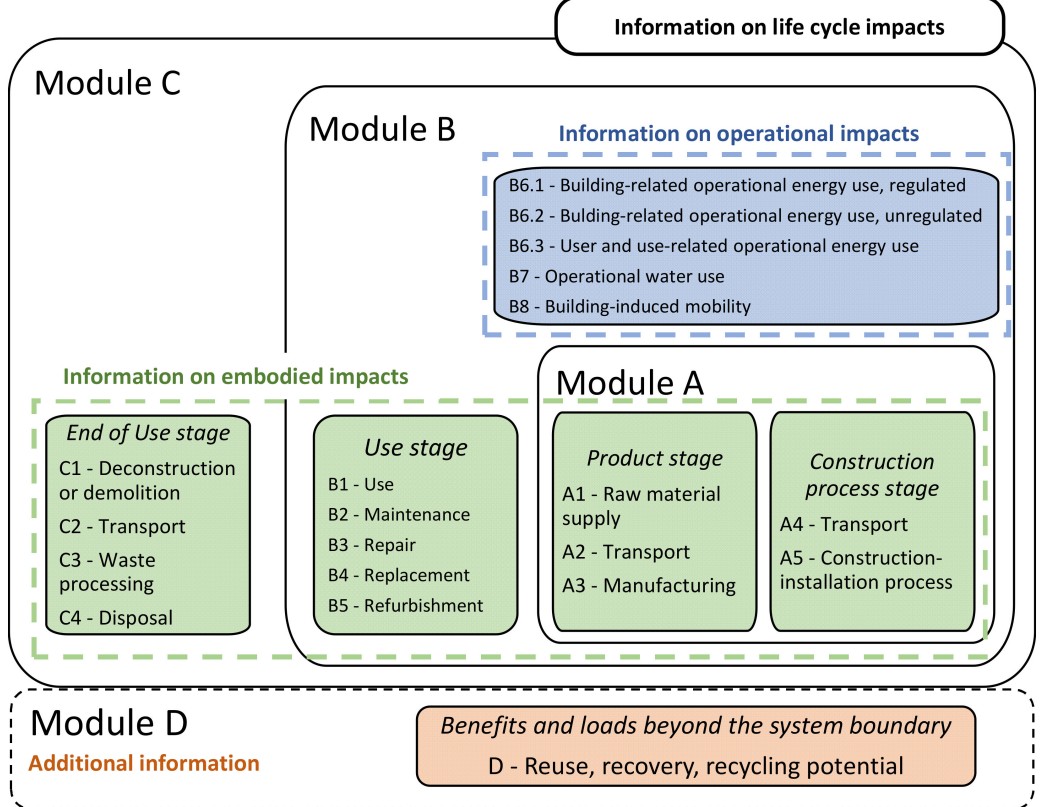

**Figure 4.** Modular framework of the system boundaries for a building's life-cycle impact. Adapted from [27] (open access article distributed under the terms of the Creative Commons CC-BY license), which was originally modified from EN 15978 and Lutzkendorf (2019) [66].

In summary, it is critical that net-zero buildings be considered in terms of an LCA of a building's operational and embodied energy using the metric of GHG emissions, which will both encourage resource use reduction and protect the climate [27].

Payback Method (Energy)

The primary emissions factor was identified as being the most significant factor in the life-cycle assessment balance of GHG emissions [59]. As this factor was reduced, net zero became increasingly difficult to achieve. In the definitions of NZEB where the source metric is used, the quality or weighting factor of the imported primary energy is assessed, and this factor is used to calculate the amount of energy that the on-site renewable generation has avoided from being imported [58]. In other words, the worth of the renewable energy which is exported back to the grid is found by calculating the avoided primary energy from the grid using the primary energy factors of the grid itself. Hence, as the grid becomes more efficient (renewable), the worth of the exported renewable energy becomes less.

In the payback method proposed by [58], the total primary energy embodied within the energy imports, and within both the building and the renewable generation system, are added together. The renewable energy exports are subtracted from this giving an energy

balance. If the balance is zero, then the building is a net-zero energy building. However, the conclusion to be drawn from this payback method, and which also applies to the other offset methods, is that offsetting the embodied energy with on-site renewable generation becomes irrelevant once the grid (primary energy) becomes 100% renewable.

Transportation and Electric Vehicles

Transportation needs are influenced by the building location relative to user needs. The transportation of a building's occupants is not generally included in any energy/emissions analysis due to the difficulty in determining comparable boundaries; however, emissions can be significant (in the USA, it is around 20% of primary energy) [27,60]. Currently only Switzerland and Norway attempt to include mobility emissions into the LCA of a building; however, this will be included as a user's activities module in a future EN 15643 standard [27].

On the positive side, electric vehicles (EVs) are starting to be used to contribute to a building's energy mix through energy storage and re-use at times when renewables (potentially) are not available. This is through a Vehicle 2 Home (V2H) connection and requires an intelligent management system [67]. The inclusion of EVs within the system boundaries of a building could assist the building to become a net-zero-energy or GHG-emissions building, depending on the GHG emissions LCA of the EV.

Water

Water can be included as part of the source energy considerations; however, it is unclear whether those articles using a source energy definition; subsequently, equating this to primary energy actually take this into account, as water is considered secondary, not primary, energy [9]. Although only using a site energy definition, both fresh and wastewater flows have been included in the system boundaries of a zero-emissions building model assessment tool [68]; however, the management of resource flows of water and biomass are not generally considered in assessments of the environmental impact of a building. The energy consumption and resulting GHG emissions from fresh drinking water can be included in the LCA of a building [27], but water emissions are more likely to be included on scales greater than the single-building scale.

Measurement and Verification (M&V)

An invaluable inclusion could be an M&V process to evaluate the load match or grid interaction indices, and to check the indoor environmental quality (IEQ) [46,47]. Verifying the actual energy flows would validate the achievement of net-zero policy targets. Recommendations are for an on-site monitoring system using hourly GHG emission factors for energy sources as well as the use of marginal electricity emission factors (these represent the marginal changes in the GHG emissions caused by variations in the non-baseload generation), which better represent the localised quality of the energy mix [27]. Similarly, the suggested use of outcome-based building codes would assess the actual energy performance of the building and could be compared with a compliance standard based on codes associated with either prescriptive requirements or performance modelling.

While NZEB can be achieved if one were to have no heating or cooling, the measurement of IEQ will determine the liveability and productivity of the building space. Air change control via mechanical ventilation is necessary to avoid overheating but still provides maximum thermal comfort [46]. IEQ is a critical factor missing from most NZEB criteria. Although it is generally recognised that design should follow a PassivHaus or equivalent design philosophy for maintaining indoor thermal comfort [42,57], there is no mention of M&V for IEQ [47].

Occupant Practices and Behaviour

Another major influence on energy balance and on the achievement of net-zero design goals is the practices and behaviour of the building occupants [53]. Houses designed to allow occupants to manage their own comfort can result in a more efficient use of energy, e.g., through temporal control of natural ventilation [69]. Total energy use is related to individual practices (types of regular activities undertaken such as cooking, washing, cleaning), coupled with the number and type of energy appliances in the home [70]. Buildings

involve socio-technical interactions, and reliance on technical solutions alone is not sufficient to achieve net zero targets as occupants may not use carbon-saving technologies correctly, leading to sub-optimal operation [39]. An operations manual could be provided to householders to achieve optimal building performance [69]. Other options include the introduction of a digital building logbook to record all relevant building data including but not limited to plans, construction materials, and performance data [71], or a serious social online game such as Green My Space to educate building occupants in energy efficiency [72]. Net-zero policies do not adequately address human practices and behaviour [39].

Occupant Health

The impact of four sustainable design rating schemes (LEED, Net Zero Energy Building, Passive House, and Living Building Challenge) on human health was assessed [63]. All the four life-cycle rating schemes studied by Hu found that these buildings had a higher potential to cause negative human health impacts than a building based on the ASHRAE standard. In terms of human health particulate potential, the Net Zero Energy Building Certification could, for example, potentially produce a 1.5 times higher impact than a baseline ASHRAE building. They state that the study was limited by the choice of only one prototype building; however, the results indicate that sustainable design rating schemes need to incorporate their effects on human health.

Exergy

A new net-zero concept for buildings conceptualizing exergy instead of energy was defined by [54], where exergy is defined as the weighted energy and material flows based on their potential to produce useful work. They considered a net-zero exergoeconomic building (NZXCB), where the total annual exergoeconomic cost of the renewable-based energy exported from the building system is equal to the total annual exergoeconomic cost of the network energy delivered to the system. Ahmadi [54] demonstrated that the NZXCB concept was the most promising out of a number of net-zero metrics tested (such as net-zero energy site, source, costs, and emissions buildings, net-zero exergy buildings, and net-zero exergoenvironmental buildings) because of its reasonable response to the variations in grid/PV parameters, sound sensitivity to the energy consumption pattern, satisfactory technical feasibility, acceptable economic profitability and plausible annual environmental impact prevention. They suggest that it could be used to complement existing net-zero building concepts.

Regenerative Paradigm

While the concept of a NZEB may not be enough to solve the economic and ecological crises [29], buildings might be designed and operated using a regenerative design such that they can add value rather than just have a net-zero impact on the environment [73]. This paradigm would use a life-cycle analysis based on the maximum energy efficiency coupled with a renewable-dominated energy mix and using a circular economy framework for materials [29].

The circular economy extends the cradle-to-gate definition given in Figure 3 and is represented in Figure 4 (and Figure 3) by Section D, which covers the re-use, recovery, and recycling potential of the building materials at end-of-life. Currently, Section D is considered to be beyond an LCA system boundary; however, this must be included in a circular paradigm. Future research should focus on holistically accounting for embodied energy in both renewable energy systems and in manufactured building materials, as well as including regenerative energy strategy buildings creating a sharing economy, which would be necessary for scaling up the net-zero concept to a district level [3].

Policy

In the USA and the EU, nearly-zero and net-zero energy buildings have been written into the legislation and are the main driver of this market [32]. Similarly, national and local policy should include net-zero GHG emissions as the next and more ambitious goal (compared with building energy performance alone), along with voluntary building certification schemes to increase the proportion of net-zero GHG-emissions buildings in the building stock [27].

Social factors can influence the technical solutions proposed for achieving zero-carbon buildings [39]. They state that "inclusion of human factors in policies must be addressed as there is a significant gap between policy intentions and actual practices", and that energy use due to user appliances (e.g., TVs, computers, etc.) is increasing and should be considered in future definitions and policies.

A framework to provide a clear description of system boundary conditions for a life-cycle energy assessment (LCEA) has been proposed [62]. They conclude that integrated policies across the three areas of regulation, information, and incentives were required to promote best practices in constructing low-life-cycle-energy buildings.

Good policy is imperative for the achievement of net-zero life-cycle primary energy and GHG emissions, and three policy actions are suggested [59]:

1.  Integrate life-cycle-embodied environmental flows into building performance regulation and certification, as the decreasing greenhouse gas emissions intensity of electricity grids can result in the initial embodied greenhouse gas emissions never being displaced through onsite electricity generation. They suggest the use of subsidies for low embodied environmental flow materials or performances.
2.  Batteries can be around 50% of the total cost of a solar PV system. Where they are not needed, net zero could be financially viable within a reasonable time (4 years in the given case). The provision of a smart grid which enables electricity to be sold back to the grid can allow this to occur. Where grid storage is possible, it should be used, but if not, and batteries are indispensable, then subsidies should be offered to support their uptake.
3.  To embrace the complexity of life-cycle environmental performance, urban-planning policy needs to be based on science and evidence. In particular, complexity exists in the relationship between the GHG emissions intensity of the electricity grid, building typology (height and on-site renewables generation potential) and the achievement of net-zero life-cycle primary energy and GHG emissions buildings. This is essential for planning future net-zero neighbourhoods.

Governance suggestions for achieving zero-energy building targets are as follows [48]:

1.  A clear, stable, long-term, collective target is the basis for governance, e.g., to push the boundary of ZEBs beyond the individual building level.
2.  Key building regulations for energy code compliance, e.g., policies related to capacity-building, education, benchmarking, and verification and code compliance checks, and how they can be implemented effectively at the local level.
3.  Key governing strategies, e.g., the complexity of ZEB targets requires broader participation from a range of societal actors who will directly influence the policy outcome, including influential local officials, developers, consumers, building owners and occupants.

The suggestion from point 3 above was successfully undertaken in the development of a zero-carbon-homes policy through the establishment of a ZeroCarbonHub which invited participation from a wide range of stakeholders [74]. This moved the policy development process away from a ministerial level to one with a more visible and public profile. Stakeholders ranged from campaigning groups (e.g., WWF), independent professional groups (e.g., Royal Institute of British Architects), the Local Government Association, planners, academics, and green building groups (e.g., PassivHaus Trust), as well as the more traditional industry representatives (e.g., house builders, product manufacturers). There was also consultation with other professional and trade groups encompassing energy, manufacturing, and building, and with government ministers across different departments. The conclusion was that rather than presenting stakeholders with a fixed set of policy options, the best outcome was to get these stakeholders to engage in the specifications of these options and in the process of policy making. This allowed policy to become a product of negotiation which promoted collective learning and was a powerful model for the design and legalization of net zero policies. The result was a policy which was more

realistic, in that it combined significant carbon reductions whilst also being sensitive to commercial constraints.

Net-Zero Standard

Net-zero standards have only been discussed in terms of building standards. The following questions were posited regarding whether ASHRAE should pursue net-zero energy as a standard [75]:

1.  How far back up the energy chain should the definition reach?
2.  Would the standard be design-oriented or performance-oriented? If both, perhaps it could use a tiered system, e.g., Tier 1 is the model and design, whilst Tier 2 is the operations, maintenance, and usage.
3.  If a standard, what are the legal implications? For example, if the building energy usage does not actually fall to net zero, can the owner sue?
4.  What happens if the building is not on pace to meet its goals as the designated net zero timeframe approaches? Would the owner sacrifice occupant comfort in order to comply?
5.  Should the real-estate market establish a consistent definition as they work directly with building clients?
6.  Should the standard focus on the key issue of climate change, i.e., carbon emissions, or on the proxy of energy use [4]?

If the energy supply becomes decarbonised faster than the building manufacturing and materials, which is likely to be the case given the uptake of renewables, then the embodied carbon would become the more important issue for compliance.

In terms of building design, the use of a zero-carbon building standard (determined by carbon emissions based on the primary energy use) would allow for a more varied design space of up to twice the size compared with that achievable using an energy metric [4].

Systems Approach

By changing the system boundaries, a shift away from NZEB and towards net zero energy communities, neighbourhoods or districts begins. This shift is apparent in the definition given by the California Public Utilities Commission, 2007 (referenced in [73]) where they refer to Zero Net Energy as "no net purchases from the electricity or gas grid, at the level of a single project seeking development entitlements and building code permits", where a project could be defined as a single building or a whole development.

There is a need to expand the net-zero GHG emissions buildings definition to include neighbourhoods and cities to account for those buildings which are unable to achieve net-zero GHG emission levels alone [27].

*Unintended Consequences*

The potential of buildings to meet NZEB criteria is dependent on their floor area with respect to their height, with increased heights decreasing the chances of them reaching net-zero energy [59]. This is due to the solar generation capacity being limited by roof area, whilst the building requirements associated with reduced daylighting and increased plug loads relative to heating and cooling increase with height (from studies mentioned in [73]: Griffith et al. 2007 [52], and Goldstein, Burt, Horner, and Zigelbaum (2010)). These studies suggest that buildings below three storeys are most suitable for NZEB, and that urban density at this level would increase the amount of energy required for transportation. They also suggest that having a requirement of renewables generated on-site would limit the roof space available for gardens, food production, water collection or simply personal open space. There must also be a balance between building heights and the design of the surrounding urban space, with consideration given to green space and its associated carbon offsets, health/biodiversity benefits with respect to population density and the urban density required for public transportation [59]. Goldstein et al. concludes that excluding transportation considerations, even though these are difficult to include in practice, could thus lead to sub-optimal outcomes in energy usage on the larger urban scale.

Weather variability must also be considered when writing policy for net-zero buildings. A study on achieving zero energy affordable housing using solar PV installation determined that costs were not distributed evenly across the state of Virginia (USA), in part due to differences in solar radiation and ambient temperature [76]. Different climates may not only affect the energy required to maintain occupant comfort (different heating/cooling loads, quantity/quality of insulation) but also the efficiency of some mechanical heating/cooling systems (e.g., heat pumps) and energy generation (e.g., solar PV, wind), and can affect the heating/cooling types required (e.g., insulation only, evaporative cooler, mechanical fan) [45,51]. Weather data used to inform the design of net zero buildings should be able to simulate future weather changes induced by global warming; otherwise, the net-zero status of the building may be compromised [3].

These unintended consequences present the limitations of defining net zero on a building scale where the system boundaries restrict the interaction of the building with other entities. This limits the flow of resources to and from the energy/water/waste providers only, and the renewable generation capacity to the roof area or a nearby generation site.

Hence, the net-zero concept should encompass more than just single buildings in isolation. It should consist of a systems approach where buildings are recognised as being just one part of an interconnected system—much like an ecosystem—and where each part of the system works more effectively through synergistic interactions, e.g., a shortfall in co-located renewable generation could be supplemented by renewable energy generated on a nearby building with a larger roof space, able to produce more than its own requirements.

The following section discusses the definition of net zero at a community scale.

### 4.3.2. Community Scale

A community is defined here as more than one building of the same typology (e.g., typically all residential housing, or all apartments) [28]. This comprises occupants and their interactions, and the services within the system such as electricity, gas, water, and waste.

Net-zero terms used for this scale include:

| | |
|---|---|
| Net-zero emission community | Net-zero community |
| Net-zero energy neighbourhood | Net-zero energy district |

Note that the term 'community' is interchangeable with communities.

There were three articles (6% of the 53) relevant to this scale.

No clear definitions of a net-zero energy/emissions community were found in a review of the literature of net zero on the community scale [28]. This was due to: a lack of defined physical system boundaries, a lack of agreement on which metric to use for balance calculation and evaluation, the varying time scales used (monthly, annual or life cycle), and the different types of balance used (import/export, load/generation) [28].

In one article modelling of a net-zero neighbourhood unit, consisting of eight two-storey buildings with two types of units (two and three bedrooms) on each floor was performed [45], but only investigated the net energy of the complex as a whole and neglected to discuss any of the potential systems benefits through energy interactions between the units, i.e., energy sharing. These blocks of units will thus perform in a similar way to a single NZEB with multiple occupants. No definition of a net-zero community was proffered.

Another article discussed the efficiency benefits of community heating and cooling systems where the use of cogeneration and/or trigeneration is critical to economic, environmental and energy savings [77]. They state that the NZEB concept may be successfully extended to create a net-zero energy city if there is a closed loop between the energy, exergy, resources, and waste of a community. Advantages can be taken from the diverse energy requirements in a community, along with other system benefits such as higher efficiency, lower peaks, energy storage and economies of scale [28].

A net-zero-emission community (NZEC) has been defined as net-zero operational $CO_2$ emissions at a community-wide level as a whole [28]. They suggested the use of an

energy master planning (EMP) framework rather than rating schemes or guidelines, to better manage the planning and design, technical solutions, and assessment of the energy system performance, and the use of an emissions metric as it is aligned with international objectives and is easy to assess [28].

*Definition*

The definition of a NZEB forms the basis for the following suggested definition of a net zero energy/emissions community:

- More than one co-located building of the same typology [28].
- A single entity owns and/or oversees the energy operations [28].
- Buildings are energy efficient (e.g., PassivHaus standard or equivalent) [35].
- Not all buildings within a net-zero community are required to be an NZEB [28].
- Renewable electricity and thermal generations units are both internally (between buildings/apartments) and externally (grid) connected.
- The use of an energy management system or plan to optimise resource use [28].
- Includes operational energy/emissions only [28].
- The annual balance of operational $CO_2$ emissions is zero across the community as a whole [28,54].

*Future*

The definition of a net-zero community should be extended to include an LCA which can encompass all environmental impacts and specifically, or at least, the impacts of embodied energy [28].

Design

Designing a community with energy use in mind in the initial design stages presents opportunities such as the integration of community heating and cooling and the use of renewable thermal energy and waste heat sources. The use of an energy master planning (EMP) framework can help to achieve this [28]. The community could thus benefit from the use of cogeneration (combined heat and power) or trigeneration (combined heat, cooling and power) energy, which is more efficient on this scale and offers considerable economic, environmental and energy savings [77].

The design process should also consider the social and ethical values of the building's occupants. There are numerous sustainability benefits to a net-zero community; hence, social metrics and energy resilience should also be taken into account in either or both of the design or regulatory spaces [28].

Policy

There does not appear to be any regulatory or policy requirements in the literature around net-zero communities. To reduce emissions on this larger scale, policy will need to be developed, and there is a will in the international community to accomplish this [28]. In terms of implementing an EMP, three universities in Melbourne (Australia) have their own EMP's and sustainability targets [28]. These are the University of Melbourne, RMIT University, and Monash University, which has also established a Net Zero Universities Initiative.

Transportation

Transportation emissions are crucial to consider as operational emissions alone have been found by Lausselet et al., 2019 (as referenced in [27]) to contribute up to 15% of the total GHG emissions in the life cycle of a net-zero GHG emissions neighbourhood. As the uptake of EVs increases, these should be considered as part of the energy balance when they are being charged within the community [28].

### 4.3.3. Urban-System Scale

The urban-system scale consists of the interactions of a mixture of different building typologies, e.g., commercial and industrial building(s), government building(s), and residential housing. It comprises occupants and their interactions, the resources and services within the system such as electricity, gas, water, and waste, and can include the environment in which the system resides, and the transportation needs of the occupants.

Net-zero terms used for this scale include:

| | |
|---|---|
| Net-positive energy | Net-zero carbon footprint |
| Net-zero energy | Net-zero energy city |
| Net-zero energy community | Net-zero multi energy system |
| Net-zero energy district | Zero-net energy community |
| Net-zero carbon city | |

There were eight articles (15% of the 53) relevant to this scale.

The wide variety of terms used for this scale is an indication of the variety in choice of system boundaries, the included metrics, and the interactions. Definitions from the articles on this scale could be separated into two main categories: those that consider operational energy only, and those that examine life-cycle emissions.

*Operational Energy*

Five of the eight articles fell into this category. There were some common themes:

- Very efficient (or PassivHaus) building designs, e.g., [55,78].
- Reliance on excess renewable energy generated by some buildings or nearby sites being exchanged with buildings which were only nearly zero energy—taking advantage of differences in temporal energy loads and generating abilities across the different building typologies (residential, commercial, etc.), e.g., [34,79].
- Use of energy management systems or smart grids in a systems approach, e.g., [79].
- Does not include embodied energy.
- Use a one-year timeframe [78].

There is the potential for excess renewable energy generation to cause instability in the national grid (specifically, solar PV in the middle of the day) [9,47]. One of the only articles to address this in their definition of a net zero energy district considers the integration of renewable production in the design process through the use of ratios [78]. They evaluate ratios of on-site renewable use to production, and on-site renewable use to total building electricity demand and use these to reduce the impact on the grid. To reduce the potential for grid instability, urban systems should be designed to manage energy balancing across sites which can produce and supply energy, to sites which require energy in a net zero energy approach [34].

In addition to the above themes, energy storage was identified as important by one article [79]. Large amounts of generated renewable electricity could be stored as thermal as well as electrical energy. This energy could be distributed through a comprehensive energy management system to meet the electrical, thermal and transportation needs of the community in a net zero multi energy system. Closing the energy, exergy, resource and waste loop extends the idea of this multi-energy system and introduces the idea of converting waste to fuel in a net zero energy city [77].

*Life Cycle Assessment*

Only three of the eight articles fell into this category. As discussed for buildings, an LCA of an urban system is a more comprehensive approach to emissions than considering operational emissions alone. The difficulties lie in the choice of system boundaries and the need for careful apportionment of emissions to reduce the occurrence of double accounting.

For example, choices were required when studying the net-zero carbon footprint of a mine site village [38]. The temporal range chosen was from a clear ground site through to end of life (between 5 and 20 years), whilst the emissions due to the embedded energy in the construction materials, transportation of building materials and supplies, operational energy, workers flying in and flying out, and storage, pumping and treatment of all water and waste were included. In another example within a city, extending the consideration of emissions to include transportation, landscape, infrastructure and services, as well as shifting the focus from energy to exergy, is also possible under an LCA inventory [73]. However, a careful choice of boundaries and scopes is warranted.

In the context of defining a net-zero carbon city, these difficult choices have been addressed below through clear definitions of three scopes and four boundaries, where the carbon emissions are considered to be all of the major anthropogenic emissions that contribute to climate change, i.e., GHG emissions [80]. The following section also first discusses the difference between the net-zero concept on a community scale and that of an urban-system scale and provides a descriptive definition of net zero on an urban systems scale based on the eight articles.

*Definition*

The definition of net zero on an urban-system scale is very similar to that of net zero on a community scale; however, there are a few main differences:

- Including a different mixture of building typologies on an urban scale means it may be easier to achieve net zero over shorter timeframes as energy usage times for the end users of the buildings will be different, e.g., school energy demand peaks during the day, whilst household energy demand peaks in the evening.
- Buildings which cannot achieve net zero status due to their physical (or occupational) constraints, e.g., low roof-area-to-height ratio, are compensated for by those which are net positive [73]. This could occur through the orchestration of smart grids coupled with solar energy production, which can have a stabilizing effect on the national grid [55].
- The inclusion of embodied energy on the urban scale and the potential to include emissions from transportation, food, and other goods.

A descriptive definition of the net-zero concept on an urban-system scale follows:

- Includes many buildings of different typologies, e.g., schools, housing, warehouses [28].
- Buildings are owned and/or overseen by many different stakeholders [73].
- Not all buildings within a net zero urban system are required to be a NZEB [55,78].
- There is a balance between buildings which cannot achieve NZEB status and those with excess electricity generation [34].
- Buildings are energy efficient (e.g., PassivHaus standard or equivalent) [78].
- Renewable electricity and thermal generation units are both internally (between buildings/apartments) and externally (grid) connected [73].
- The use of a smart grid, and an energy management system or plan to optimise resource use [34,79].
- The balance period is over the lifetime of the urban system for embodied emissions, whilst the operational emissions should balance to zero at least annually across the urban system as a whole [38,73].

In addition to the above generalised definition, a good definition of net-zero urban systems' scopes and boundaries has been reproduced below [80]. It is the most comprehensive one found in the studied articles. The authors discussed a net-zero carbon city and defined this in terms of three GHG emissions scopes using four system boundaries. The three scopes were:

- Scope 1—internal emissions (those occurring within the physical boundary and within scope);
- Scope 2—core external emissions (resulting from activities outside the city which include whenever the city imports a good or service that contains some embodied carbon and also falls within the activity boundary); and
- Scope 3—non-core emissions (any carbon emitting activity that falls outside the activity boundary. These emissions can occur either inside or outside the city's geographic boundary).

The four boundaries were:

1. the geographical boundaries that distinguish internal from external emissions;
2. the temporal boundaries within which emissions are tracked;

3. the activity boundary that outlines the carbon emitting activities for which a city should be held responsible and that must be accounted for in the city's carbon balance for a given scope; and

4. the life-cycle boundary that determines the degree to which the production and disposal of capital goods required for any activity are included.

Some examples of the above scopes are:

Scope 1:

- On-site power generation
- Urban transport
- Waste management

Scope 2:

- Importing electricity and water
- Commuting by public employees

Scope 3:

- Private employee commuting
- Private goods purchased outside the city
- Importing food products
- Unregulated small-scale internal GHG emissions from resident activity (e.g., barbeque)

A net-zero carbon city would be one where all Scope 1 emissions would be eliminated, Scope 2 emissions would be balanced, and Scope 3 emissions would be minimised. Table 5 presents proposed system boundaries and definitions for achieving net zero across all scales.

**Table 5.** Summary of system boundaries and definitions to achieve net zero on each scale.

| | Building Scale | | Community Scale | | Urban-System Scale | |
|---|---|---|---|---|---|---|
| | System Boundaries | Definition | System Boundaries | Definition | System Boundaries | Definition |
| Typology | To be defined. Should consider the following boundaries: geographical; temporal; activity; and life cycle. | Single (stand-alone) building. | As for Building scale. | More than one co-located building of similar typology, single entity owns/oversees the energy operations. | As for Building scale. | Mixed residential, industrial, commercial, and government building(s). |
| Net zero measurement metric | Across the whole building operational phase, embodied emissions and occupant activity system. | Scopes 1, 2, and 3 GHG emissions. | As for Building scale but balance can be achieved across the whole community. | As for Building scale. | As for Community scale. | As for Building scale. |
| Net zero timeframe | | Annual for operational and occupant activities, expected lifetime of structural materials, fixtures and fittings for embodied emissions. | | As for Building scale. | | Monthly for operational and occupant activities; over the expected lifetime of structural materials, fixtures and fittings for embodied emissions. |

**Table 5.** *Cont.*

| | Building Scale | | Community Scale | | Urban-System Scale | |
|---|---|---|---|---|---|---|
| | System Boundaries | Definition | System Boundaries | Definition | System Boundaries | Definition |
| Operational phase | Regulated (building-related operations, e.g., space heating and cooling, fixed lighting) and unregulated energy (user related operations, e.g., plug loads); fresh and wastewater flows; any biomass resources. | Energy system is grid connected, with renewable generation on-site or nearby, balance between weighted supply and demand (source energy). | As for Building scale. | As for Building scale but energy system is also internally connected (between buildings), uses trigeneration (combined heating/ cooling/power), and is controlled by an energy management system. | As for Community scale. | As for Community scale. |
| Embodied emissions (structural materials, fixtures, and fittings) | Includes materials extraction, production, transport, construction, maintenance, replacement components, decommissioning, reuse/recycling. | LCA (cradle to cradle) of all materials must be net zero or better (regenerative); LCA uses an international database of embodied environmental flows which are dynamic to capture grid greening and climate change | As for Building scale. | As for Building scale but uses a systems approach to avoid shifting emissions from one life cycle stage to another. | As for Community scale. | As for Community scale. |
| Occupant activities and socio-technical interactions | Transport to/from building for users needs, e.g., work, social, health; internal building activities | LCA (cradle to cradle) of transport component used by occupants; assessment and management of occupant behaviour and practices aiming to reduce emissions, e.g., production and use of a house operational manual, inclusion of indoor environmental quality (IEQ) control systems tailored to meet occupant needs and their technical capacity, and/or smart technology. | As for Building scale. | As for Building scale. | As for Building scale. | As for Building scale. |

**Table 5.** *Cont.*

| | Building Scale | | Community Scale | | Urban-System Scale | |
|---|---|---|---|---|---|---|
| | **System Boundaries** | **Definition** | **System Boundaries** | **Definition** | **System Boundaries** | **Definition** |
| Design | Building and fixtures; energy system; indoor environment; outdoor environment within property boundary | Building is sustainable/solar passive, e.g., Passivhaus; accessible (meets diverse needs and is future proofed for these); regenerative (adds environmental value and/or adds energy for sharing to upscale to community model); indoor environmental quality meets relevant air quality, comfort, and health guidelines/standards; outdoor environmental air quality meets relevant standards. | As for Building scale but includes between-building infrastructure. | As for Building scale but uses an energy master planning (EMP) framework to integrate heating/cooling/renewable thermal energy/waste heat sources; consider occupants' social and ethical values–social metrics and energy resilience; employs regenerative sustainable urbanism principles. | As for Community scale. | As for Community scale. |

*Future*

The definition of net zero on the urban-system scale is quite complex. Achieving net zero on this scale is beyond the capacity of a single entity and will require a multi-level and collaborative approach. It is essential that the urban system not only reaches net zero, but then has the capacity to drawdown emissions. Similar to the future issues raised regarding NZEB definitions, measurement and verification processes are crucial to effecting real emission reductions across an urban system. However, at least four areas should be examined in future research:

1. Governance
2. Design
3. Measurement and verification
4. Circular framework

These are discussed in Section 5.

## 5. Future of Net Zero Research

From this systematic literature review which focused on articles providing a definition of net zero, four areas have been identified as requiring further investigation (Section 4.3.3). These four areas are expanded upon below. To support some statements in this section, references sourced from outside of the literature review search string were included.

### 5.1. Governance

The importance of good governance cannot be overstated, as the complexity of achieving net zero and beyond requires the collaboration of multiple stakeholders. To achieve community-wide and country-wide emission reductions, governments should enact scientifically robust policies and regulations which provide a net zero target with widespread support.

### 5.1.1. Consultation, Collaboration, and Communication

A framework would enable effective consultation, collaboration, and communication based on the research into the establishment of a ZeroCarbonHub [74]. This will bring on board an engaged population and achieve effective policy outcomes which can translate into laws and regulations.

The framework should include a requirement to:

- Consult with all relevant stakeholders to bring into effect the required changes in the supply chain (from mining, manufacturing, delivery, construction, servicing, operation, and the end-of-use options of re-use and recycling).
- Collaborate within and across the four sectors of university, government, industry, and the public to foster transdisciplinary synergies in research, sharing of information, design, and implementation of net zero urban systems.
- Communicate within both the consultation and collaboration processes and include informing end users, e.g., awareness raising in the community.

### 5.1.2. Definition

As evidenced in this review, a clear definition of net zero is needed for the studied scales in the urban environment. Within these scales, the one most likely to achieve net zero is the urban-system scale due to the potential synergies of its interacting elements and energy flows. The system boundaries and the included scopes of emissions should be defined. The framework for this has been set but needs refining and standardising [80]. For example, the activity boundary in this framework could be defined in many ways, and it would be beneficial for this to be standardised internationally. Standardised system boundaries must carefully consider the apportionment of emissions to reduce the occurrence of double accounting.

The definition should also include the use of a dynamic approach [27] that will consider changes in the emissions factors of materials and transport with time due to increased renewable generation. Plug loads (user appliances or unregulated energy) have not been included consistently in international definitions of system boundaries. As the plug loads are increasing, it is imperative that their inclusion be standardised. All environmental impacts should be considered in net-zero definitions, as should transport, water and food.

One of the most pressing needs identified was for a consistent use of scientific nomenclature. Shorthand words such as carbon and emissions do not make it easy to compare and assess article findings. When discussing anthropogenic gas emissions which contribute to global warming, the use of GHG emissions or $CO_2$-e are suggested. If only carbon dioxide emissions are being considered this should be stated explicitly and $CO_2$ can be used. The use of energy needs to be clearly defined as well as its relationship to GHG emissions.

In summary, in addition to the net-zero-energy building definitions (most of which do not include embodied energy due to manufacturing, materials, construction, maintenance or end-of-use), a net-zero urban-system definition should include:

- standardised system boundaries
- standardised GHG emission scopes
- standardised terminology
- a dynamic approach
- life-cycle energy and GHG emissions from:
  - plug loads (unregulated energy)
  - transport
  - food
  - water

### 5.1.3. Policy and Regulation

There do not appear to be any regulatory or policy requirements in the literature around net-zero communities or urban systems. Good policy and regulation are imperative

for the achievement of net-zero life-cycle primary energy and GHG emissions. Policies and regulations are required to reduce both energy (to reduce resource use) and GHG emissions (to reduce climate impacts) on all scales—those focused on energy alone are not enough to achieve climate targets [59]. Net-zero standards for building have been considered but not implemented; this is probably due to the complexity around issues of metric, compliance, and accountability [4].

The following points should be considered and included in policies and regulations:

- Energy and GHG emission targets
- Capacity building
- Education
- Outcome-based building codes, which includes measurement, verification, and code compliance checks [27]
- Engagement of a wide range of stakeholders to undertake policy making using a transparent and public process [74]
- Evidence-based science for urban planning, building performance regulation and certification
- Energy resilience [28]
- Life-cycle-embodied environmental flows, as offsetting the embodied energy with on-site renewable generation becomes impossible once the grid (primary energy) becomes 100% renewable [59]. This could include the use of GHG emission budgets or materials.
- Incentives, e.g., subsidies for renewables storage and for low embodied environmental flow materials or performances [59,62]
- Human behaviour and practices to address the gap between policy intentions and actual practices [39]
- Consideration of different climate/weather zones
- The potential to extend a net zero concept to one where existing GHG emissions are removed (drawdown).

5.1.4. Continuous Improvement

Engineers and companies often use continuous-improvement models or loops which involve identifying opportunities, planning for improvement, executing the change, measuring the results, and then returns to identifying opportunities [81]. Similarly, to optimise the successful achievement of net-zero GHG emissions on any scale, there must be a feedback loop to inform policy and regulation. Feedback would be based on real-world data measurement and must be in a form suitable to support decision making processes. This would help to ensure results from the design and M&V processes continue to be meaningful and are responsive to the changes required to achieve emission reductions.

*5.2. Design*

The design of buildings, new estates, urban spaces, and city infrastructure should be controlled by the definition of the net-zero concepts in this review, which have demonstrated the importance of a systems approach. To design for this systems approach on an urban-system scale will expand the usual energy and economic exchanges with power utilities to include collaboration with many other stakeholders [73]. Designing for net zero is important for new builds but equally important for retrofits or renovations. Renovations present opportunities for introducing structures, materials, or processes as part of an interacting system which underpins the net-zero concept.

Articles on urban design for net zero were not found in the 53 definition filtered results. A search of "urban design" in the TAK of the original 1569 net-zero articles returned only four articles. Although these did not provide a definition, they discussed the following points:

- The use of environmental urban design guidelines to design a residential community, where the guidelines include homes highly rated for thermal comfort, gas-boosted

solar hot water, solar PV, high energy star rated appliances, ceiling fans, treated stormwater used for toilets, laundry, and irrigation, and an in-home energy-use feedback display [70].

- The assessment of water as part of a comprehensive urban design strategy [82].
- An integrated approach to urban design and the use of water resources [83].
- Exploring the economic aspects of distributed energy resources in sustainable urban design [84].

Based on these points, the inclusion of water in a net-zero urban system design is deemed essential.

The following sections detail some urban design considerations for achieving net-zero GHG emissions.

### 5.2.1. Dynamic Variables

In practice, an urban system is undergoing constant change involving both expansion (new designs for new spaces), and retrofitting or re-design. Research is needed to investigate changes in the following features/variables:

- Embodied environmental flows coefficients, or emissions factors of materials and transport change with time as energy systems become more renewable.
- Forecasting of weather data for the future global-warming-induced climate changes.

### 5.2.2. Energy Management System

A comprehensive energy management system is required to meet the complex and interconnected electrical, thermal and transportation needs in an urban system. For example, the use of energy master planning (EMP) as implemented at the three universities in Melbourne, Australia to prioritise energy flows to capitalise on co- and tri- generation [28].

The ever-increasing use of renewable energy generation will begin to cause problems with national grid stability unless it is managed effectively through energy sharing and storage within communities or on a larger scale. Communities designed with this in mind and with the use of EMP can reduce their impact on the grid [28].

Similarly, including battery electric vehicles as a part of the electricity storage solutions for homes or office buildings will require smart management systems [28], especially when this concept is envisaged to be extended to include larger battery electric transport options such as buses, trams or trains on an urban-system scale.

Optimisation of energy management systems will be required on an urban-system scale.

### 5.2.3. Socio-Technical Interactions

In an urban system, there are many interactions between people, technology, and technical systems. The achievement of net zero will depend on the effectiveness of social and technical processes interactions [39]. For example, the operation of a home energy management system may not be used to optimal efficiency if the users are not well trained in how to use the software. Design criteria should be expanded to recognise this socio-technical interface, and designs should incorporate these interaction processes over the lifetime of the urban space [73].

### 5.2.4. Human Practices and Behaviour

Human practices and behaviour affect the efficiency of the use of energy and resources [53]. Design which caters for practices and behaviour will lead to better emission reduction outcomes. The suggestion of providing building occupants with operations manuals [69] needs further investigation, but it is likely that whilst a manual may educate occupants, it is unlikely to change behaviour which is bound up in a system of practice [85]. Regardless, urban design should be expanded to accommodate for the human factor with specific reference to emissions reduction.

### 5.2.5. Environmental Impact

Urban design has a direct impact on the environment through its ability to regulate energy and resource flows, and through its physical footprint. All environmental impacts such as emissions to air, surface waters, groundwater, and soil, consumption of resources, and production of wastes [68] should be considered when designing for net zero.

As well as the outdoor impacts, urban design must consider indoor environmental quality. This means reducing the impacts of buildings and spaces on heat, natural light, air quality, sound quality (where outdoor noise may contribute negatively to indoor sound levels), and the inclusion of IEQ control systems tailored to meet occupant needs and their technical capacity to operate the systems [46].

### 5.2.6. Regenerative Sustainable Urbanism

A new approach to help reverse the negative ecological footprint of cities is regenerative sustainable urbanism, which combines biophilic design (design to reconnect humans to nature) and regenerative development and design (design to promote regeneration of living systems) [86]. Although this concept did not appear in the net-zero search, it is a crucial element to consider if net zero is to be achieved. Regenerative sustainable urbanism builds back social and natural capital through the integration of green infrastructure into the built environment and urban spaces and the restoration of degraded urban and regional ecosystems [86]. As part of a suite of other benefits, regenerative sustainable urbanism can help regulate climate through the mitigation of GHG emissions [86].

### 5.2.7. Unintended Consequences

The design criteria for urban systems must carefully consider all aspects and implications of the system, particularly any unintended consequences. Some of these have been identified and require further investigation:

- There are competing interests for limited roof areas, e.g., roof space which is used for renewables becomes unavailable for human recreation, plants, or food production.
- Although high density living may promote efficiencies in energy savings through solar energy sharing, the roof space on a high-density apartment of multiple storeys does not provide enough solar generation capacity to cater for the needs of the occupants.
- There must also be a balance between building heights and the design of the surrounding urban space, with considerations given to green space and its associated carbon offsets and health/biodiversity benefits versus population density versus the urban density required for public transportation.

### 5.3. Measurement and Verification

Measurement and verification (M&V) should be prioritised and be regulated for, to ascertain the effectiveness of any net-zero policy or regulation. Without this, there is no comparison between expected and actual performance, no accountability, and net-zero targets are unlikely to be achieved. For buildings, the use of outcome-based building codes, as mentioned previously, will assist in driving the M&V approach, and the inclusion of certification will allow the standardised communication of results.

Australian rating schemes such as the National Australian Built Environment Rating System (NABERS) [87] and the Nationwide House Energy Rating Scheme (NatHERS) [88] are beginning to include M&V in terms of single buildings. For example, NABERS currently measures and rates office buildings, shopping centres, hotels, and the common use areas of residential apartment buildings for energy, indoor environment, water and waste. However, they do not assess residential homes. Residential energy will be assessed under a new NatHERS Home assessment rating scheme to be introduced in 2022, and these are climate zone sensitive, but do not include measurement of other variables. More work needs to be carried out on this process and to extend it to the complexity of urban systems. Further research can identify how and by whom the cost burden of compliance will be borne.

### 5.3.1. GHG Emissions

A M&V process at its simplest could be used to assess the import/export balance of energy or GHG emissions of a NZEB [47]. However, the practicalities of this are still complex and involve metering or sub-metering depending on the chosen system boundaries, inclusion of a temporal factor, adjustment for external weather conditions, and consideration of occupancy rate, activity, and comfort levels [47].

In terms of embodied GHG emissions, M&V becomes even more complicated due to the use of a wide variety of materials resulting in many stakeholders, supply chains and processes, not to mention the different interpretations of an LCA [6]. This is an emerging field of research.

### 5.3.2. Indoor Environmental Quality and Human Health

Future directions for a definition of NZEB have mentioned the inclusion of indoor environment quality (IEQ) factors [47]. If the indoor environment is uncomfortable, then the building may provide reduced liveability and productiveness, which in turn affects the NZEB status. This creates the potential for people to generate more emissions to improve their space, e.g., through the increased use of air conditioning.

IEQ can also affect human health through higher levels of $CO_2$ and VOCs (due to low air change rates per hour), exposure to ambient pollution such as particulates (from cooking, vacuuming or photocopiers), or through outdoor air penetration where there are high air change rates in areas with high levels of outdoor air pollution. NZEB assessments should include M&V of IEQ, and research opportunities exist to determine how to achieve this and what to measure, and to identify the standards required. Currently, in Australia, unlike for outdoor air, there are no legislated indoor air standards [89].

### 5.3.3. Environmental Impact

M&V of the environmental impacts of a net-zero urban system will be difficult to achieve and will require detailed identification of parameters to be measured, standards to be met, measurement periods, location of sources for accountability, and some way of certifying for all of these.

### 5.4. Circular Framework

The concept of end-of-use processes such as reuse and recycling have not been addressed satisfactorily in any of the articles in this review. Definitions of net-zero processes should be based on an LCA, and a circular framework must be included if true net-zero is to be achieved on a planetary scale. The following points should be considered:

- Standardizing an LCA approach to materials through the development of an international database of embodied environmental flows coefficients for materials [59].
- Upfront design of buildings and infrastructure for end-of use processes through careful consideration of material use, ease of disassembly, deconstruction, and resilience and the potential use of a newly developed circular economy index [90].
- Circular approach to exergy flow, e.g., closing the energy, exergy, resource and waste loop through using tri-generation (combined heating, cooling and power systems) and converting waste to fuel [77].
- Choice of system boundaries for circularity: how can these be defined on the larger urban-system scale? Or should they be defined using different metrics such as by resource or service, e.g., building materials, vehicles, construction, water?
- Does the choice of these system boundaries impact on the efficiency of achieving a circular system?

The points above are based on material and energy flows and do not include societal effects such as impacts on well-being and equity. These should be considered in a true circular framework for net zero, as the human impact factor cannot be ignored.

Table 6 presents a framework for the above four areas—governance, design, M&V, and circular framework—highlighting considerations for achieving net-zero GHG emissions across all scales.

**Table 6.** Framework of considerations within the four areas required to achieve net zero.

| Areas | Considerations |
|---|---|
| **Governance** | |
| Consultation, collaboration, communication | Consultation with all relevant stakeholders; collaboration within and across university, government, industry, and the community; communication within consultation and collaboration processes and informing end users, e.g., community. |
| Definition of net zero | Defines a clear and consistent target and timeframe with interim targets. Defines the system boundaries. Metric is GHG emissions using an LCA (cradle to cradle). |
| Policy and regulation | Policy determined through wide stakeholder participation. Must address all four areas of governance, design, measurement and verification, and circular framework. |
| Continuous improvement | Establish a system to identify opportunities, plan for improvement, execute the change, measure the results and feedback into identifying opportunities across all governance, design, M&V, and circular framework areas (continuous improvement model). |
| Unintended consequences | Across all areas, negative unintended consequences should be considered and avoided if possible. |
| **Design** | |
| Building | Must be sustainable, e.g., Passivhaus. Has a lifetime GHG emissions budget. Is accessible (meets diverse needs and is future proofed for these), regenerative (adds environmental value, adds energy for sharing to upscale to community model), and considers occupant health. |
| Dynamic variables | Considers changing environments, e.g., expansion/retrofit requirements and climate change, and changing parameters, e.g., emissions factors and embodied environmental flows. |
| Energy management system | Optimises interconnected electrical, thermal and transportation needs. |
| Socio-technical interactions | Recognises the socio-technical interface, incorporating lifetime interaction processes, and considers education/training/smart technology possibilities to minimise emissions. |
| Human practices and behaviour | Minimises unnecessary emissions caused by occupant behaviour and systems of practice. |
| Environmental impact | Minimises emissions to air, surface waters, groundwater, and soil, consumption of resources, and production of wastes. Considers indoor environmental quality through reducing impacts of buildings and spaces on heat, natural light, air quality. |
| Regenerative sustainable urbanism | Helps to regulate the climate through mitigation of GHG emissions and builds back natural and social capital through the integration of green infrastructure. |
| **Measurement and Verification (M&V)** | |
| GHG emissions | Undertake an LCA (cradle to cradle) of both operational and embodied emissions. |
| Indoor environmental quality, human health | Assess and minimise exposure to heat, air pollution, e.g., $CO_2$ and particulates. |
| Environmental impact | Determine parameters to measure, standards to be met, measurement periods, location of sources for accountability, and a process for certification. |
| Timeframe | Assess the above M&V parameters before and during build, during operation (scheduled), and for the deconstruction/reuse phase. |
| **Circular framework** | |
| Standardisation | Standardising an LCA (cradle-to-cradle) approach to materials, exergy flows, and system boundaries. |
| Design | Upfront design of buildings and infrastructure for end-of use processes. |
| Environmental and societal impacts | An LCA approach should be taken to minimise impacts on the environment and on society such as well-being and equity. |

## 6. Conclusions

This literature review set out to investigate the history and background of net zero and to determined how the many different net-zero terms have been defined in the academic literature. From these definitions, knowledge and process gaps were identified which indicate opportunities for future research.

The first climate-related use was of zero-net $CO_2$ emissions in 1991. Since then, although it was technically used originally with regard to GHG emissions, the net-zero term has been used across many different areas and with many different meanings. In fact, just over half of the reviewed articles that mentioned 'net zero' or 'zero net' did not

mention the word 'emission' anywhere in the article. The most commonly used term was 'net-zero energy building'. Hence, researchers searching for articles on net-zero emissions must be aware of this point.

A number of research opportunities necessary for achieving global net-zero GHG emissions were revealed through the analysis of the net-zero definitions, the most pressing being a clear need for a governance framework which includes:

- Consultation, collaboration, and communication within and across all sectors and stakeholders to ensure endorsement (buy-in), equity, and effectiveness, in decision making, research, and implementation.
- Setting ambitious targets in an equitable manner.
- The use of scientifically consistent nomenclature.
- A clear definition of net zero on the appropriate scale, e.g., building scale or urban-system scale, which includes explicit system boundaries and emission scopes, life cycle energy and GHG emissions, and uses a dynamic approach.
- A circular framework which considers not only material and energy flows but also societal effects such as impacts on well-being and equity.
- Policies and regulations to reduce both energy (to reduce resource use) and GHG emissions (to reduce the climate impacts) on all scales.

Other research opportunities to help achieve net zero are:

- Net-zero targeted urban design to promote energy and resource synergies, reduce environmental impact, and to consider human practices and behaviour, and socio-technical interactions, with care taken to minimise unintended consequences.
- Measurement and verification (M&V) using GHG emissions as a metric, but also establishing M&V of indoor environmental quality (IEQ), human health, and environmental impacts as part of the net-zero approach.
- Continuous improvement (or feedback) in governance, design, and M&V.

In general, the net-zero literature is limited in the area of social science, a field critical to the achievement of net-zero targets.

"The area of adequacy and fairness is ripe for work across multiple disciplines. Setting net-zero targets at a country or company level cannot be done only by natural scientists or economists. Ethicists and social scientists are needed to help explore how fairness concepts apply to today's multinational corporations—which span multiple countries and sectors and involve staff on incomes ranging from the lowest to the highest. Equity must be made a central part of the process, including in tools used to design net-zero targets so that the wider implications of assumptions and decisions become clear." [16].

At a university level, the authors recommend the establishment of dedicated strategic and transdisciplinary initiatives on zero emissions (or similar), to draw together people from across many disciplines and make use of the synergies which will come from transdisciplinary research.

**Supplementary Materials:** The following supporting information can be downloaded at: https://www.mdpi.com/article/10.3390/su14053057/s1, List S1: Manually sourced references.

**Author Contributions:** Conceptualization, J.L., G.M.M. and D.A.M.; methodology, J.L.; formal analysis, J.L.; investigation, J.L.; data curation, J.L.; writing—original draft preparation, J.L.; writing—review and editing, J.L., G.M.M. and D.A.M.; visualization, J.L.; supervision, G.M.M. and D.A.M.; project administration, G.M.M. and D.A.M.; funding acquisition, G.M.M. All authors have read and agreed to the published version of the manuscript.

**Funding:** This research received no external funding.

**Institutional Review Board Statement:** Not applicable.

**Informed Consent Statement:** Not applicable.

**Data Availability Statement:** Not applicable.

**Conflicts of Interest:** The authors declare no conflict of interest.

## Abbreviations

| | |
|---|---|
| $CO_2$ | carbon dioxide |
| $CO_2$-e | carbon dioxide equivalent |
| EMP | energy master planning |
| EV | electric vehicle |
| GHG | greenhouse gases |
| GWP | global warming potential |
| IEQ | indoor environmental quality |
| IPCC | Intergovernmental Panel on Climate Change |
| LCA | life cycle assessment |
| M&V | measurement and verification |
| NZEB | net-zero energy building |
| PV | photovoltaic |
| SLR | systematic literature review |
| TAK | title, abstract, and keywords |

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
