# Peer review of "Identifying Knowledge and Process Gaps from a Systematic Literature Review of Net-Zero Definitions"

_sustainability, doi:10.3390/su14053057_

Round 1
Reviewer 1 Report
Overall, this paper is well written and the research paper is interesting. However, the following comments shall be addressed:
1) Abstract section is not written properly. It should be written by following steps;
- Write the concept as per your paper title (3 lines minimum).
- Objective (2 lines)
- Tools (2 lines)
- Output (3 lines)
- Output with application in real life or in the industry (mandatory)
2) In the Introduction section, the background of this research domain is way sufficient (could to too much) and the justification for this research, i.e., research gap, is rather weakly presented. Furthermore, the proposal of this research does not seem to be clearly presented to address the research gap. Please revise the Introduction section accordingly.
3) Literature review shall be comprehensive (rather than brief) to discuss the right breadth of knowledge and recent works in the area. Remove the word briefly in the Literature Review section
4) Better to add a contribution table.
5) The English writing should be improved. For academic writing, try to avoid using Maybe, And, but, etc. to start a sentence. Try to write research articles based on 3rd party writing style, hence avoid using We, our, etc
6) Output with application in real life or in the industry (mandatory) in the conclusion section.
Author Response
Thank you for taking the time to review this manuscript, the authors appreciate this.

Reviewer 2 Report
The study presents a systematic literature review about the existing knowledge on the Net Zero concept, from its origins and multiple definitions to its application and verification. The authors discuss the Net Zero scope at several scales, from a more conceptual and abstract, passing through the "building" and ending in "urban systems" and "city" scale. In this study, the authors do not limit themselves to a descriptive discourse but also analytical and interpretive. They raise several questions throughout the main text that I consider pertinent and highlight research gaps and fields worth exploring. It is a pervasive study and should be shortened if possible. However, its well-structured, well-written, and touches, as far as I my knowledge allow, almost every topic on this matter. For this reason, the content of this article presents a tangible contribution to this field, as it summarizes the current state of the art and pushes into new perspectives dealing with this concept issues, mainly the net-zero "consistent definition" undefinition. Therefore, I recommend its publication but with some adjustments that I consider "minor revisions".
If published, I will use this paper in my research context for sure.
By the end, I want to thank the authors for their effort. In my opinion, they did a great job.
- Overall comments
- Remove the recommendation part at the end of the abstract.
- I think you have too many keywords. Please merge some of them, like "CO2, carbon, emissions".
- I suggest you try to standardize the way you present the references throughout the text: "…a study by [xx]…" or "…a study by Ahmadi et al. [xx]…" you have both forms. Use the journal GoA reference style to correct this issue.
- Despite the study extent, this research has not addressed the net-zero energy buildings field in old or heritage buildings. I admit that the used Methodology might have excluded papers on this subject. So, I suggest that this issue should be referred to in a "limitations" section placed at the end of the Methodology part and not at the end of the Conclusions. Despite this fact, the non-inclusion of this topic does not invalidate the overall quality of the research produced.
- Introduction
- Well structured, concise, and clear. The research gap was identified and addressed.
- Redundant information in line 59 to 65. You are saying the same thing in six lines. This happens here and there all over the text, and I suggest you run the main body looking for redundant info.
- Methodology
- It's clear and with a step-by-step kind of approach. I understand the process and the used criteria.
- Background
- Only in line 262 do you describe what IPCC (Intergovernmental Panel on Climate Change) is, despite you refer it several times before. Run the paper and search for these kinds of issues.
- Terminology analysis
- On page 12 lines 413, 414, you should refer that the Near Zero Energy Buildings and Net Zero Energy Buildings have the same abbreviation – NZEB – which is confusing even for experienced researchers. For the sake of understanding which "NZEB" you are referring to in the following sections, I suggest you should write its full name "Net Zero Energy Buildings" instead of "NZEB". I often questioned which one you were referring to while I was reading it (e.g., p. 17, line 583, 634, and 637).
- Future of net-zero research
- Line 1280, 1281 – I suggest you might rewrite this sentence to increase its precision, as CO2 and VOC's (cooking, vacuuming, etc.) indoor concentration occur under limited ventilation values (low air change rates per hour usually below 0.6 ACH) or in areas with highly polluted outdoor air.
- Conclusions
- The conclusions are pretty solid and consistent with the evidence and arguments presented. My only concern is with the direct citation in this part (line 1357 – 1363). I would paraphrase it, but I admit it's a style issue.
Well done. Keep up the excellent work, and all the best!
Author Response

(The authors gave the same response as above.)

Reviewer 3 Report
- The comma is almost generally misused throughout the manuscript paragraphs. As well as grammar revision is required.
- meter instead of metre in line 411
- In line 535, replace "energy" instead of "emergy." It's worth noting that the fault word "emergy" appears twice in this line.
- “Those definitions” instead of “that definitions” in line 1373.
- In the table of common acronyms, all acronyms should be listed. For example, (NZEC), (EVs), (EMP), (NZXCB) and (V2H) should be included.
- A reference is required in several paragraphs, for example, the paragraph between lines 863 and 875, the paragraph from line 977 till 994, and the paragraph from line 977 till 994.
- As stated in the abstract, the objective of this research is to find a clear net-zero definition at the appropriate scale (single building or urban system scale). However, as stated in the conclusion section, the article does not provide a clear definition of net-zero. Hence the authors declare that there is a clear need for a governance framework that includes a clear definition of net-zero on the appropriate scale. As a result, I propose that the objective of this paper be rewritten in a different way.
- From line 641 to line 652, there is a paragraph about occupant practices and behaviour. Several previous studies looked at how differences in people's lifestyles from one community to the next could affect building energy use. It's something I think needs to be addressed.
- The conclusion paragraph should restate your article and summarize the key supporting ideas you discussed throughout the work, rather than restoring the historical context mentioned in the second paragraph of your manuscript's conclusion section.
- The third boundary used to define the net zero carbon city, which has been explained from line 1011 to line 1013, is an unfinished point. And it has to be completed.
- Is the section entitled “5.1.1 Consultation, collaboration, and communication” represents the author vision or has been discussed in the past literature?
Author Response

(The authors gave the same response as above.)

Reviewer 4 Report
The paper is a very interesting literature review on "net zero" expression from building to urban scale. It is maybe too long, but it provides insightful inputs for practical action on land governance, building and infrastructure design and facility management. The authors can find some specific comments directly in the attached file.

Author Response

(The authors gave the same response as above.)

Round 2
Reviewer 1 Report
Accept
Reviewer 3 Report
The authors went over the manuscript carefully and clarified all of the points raised in the first round of revision.